# Classical and Multisymplectic Schemes for Linearized KdV Equation: Numerical Results and Dispersion Analysis

Adebayo Abiodun Aderogba [1] and Appanah Rao Appadu [2,*]

1  Department of Mathematics, Obafemi Awolowo University, Ile-Ife 220282, Nigeria; aaderogba@oauife.edu.ng
2  Department of Mathematics and Applied Mathematics, Nelson Mandela University,
   Gqeberha 6031, South Africa
*  Correspondence: Rao.Appadu@mandela.ac.za

**Abstract:** We construct three finite difference methods to solve a linearized Korteweg–de-Vries (KdV) equation with advective and dispersive terms and specified initial and boundary conditions. Two numerical experiments are considered; case 1 is when the coefficient of advection is greater than the coefficient of dispersion, while case 2 is when the coefficient of dispersion is greater than the coefficient of advection. The three finite difference methods constructed include classical, multisymplectic and a modified explicit scheme. We obtain the stability region and study the consistency and dispersion properties of the various finite difference methods for the two cases. This is one of the rare papers that analyse dispersive properties of methods for dispersive partial differential equations. The performance of the schemes are gauged over short and long propagation times. Absolute and relative errors are computed at a given time at the spatial nodes used.

**Keywords:** Korteweg–de-Vries equation; classical; multisymplectic; dispersion analysis; finite difference methods

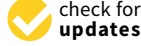

## 1. Introduction

In this paper, we solve a linearised Korteweg-de-Vries equation with specified initial and boundary conditions. The three methods include classical, multisymplectic and a modified explicit scheme adapted from Wang et al. [1]. In 1895 [2], two Dutchmen, namely Korteweg and de Vries, derived a nonlinear partial differential equation of the form

$$u_t + uu_x + \delta^2 u_{xxx} = 0, \tag{1}$$

which describes the long time asymptotic behaviour of a small but finite amplitude of one-dimensional shallow water waves. In the equation above, $u = u(x,t)$ measures the elevation (height of water above equilibrium level) at time $t$ and position $x$ while $\delta^2$ is referred to as the dispersion coefficient. There are two different mechanisms that are present, namely;

1.  Nonlinearity ($uu_x$), which tends to steepen those parts having negative slope.
2.  Dispersion, which makes dispersive wave components of different wave frequencies propagate at different velocities.

The delicate balance between these two effects leads to travelling waves of permanent form, the so-called solitary wave. It is usual to refer to the solitary wave as the single soliton solution, but when more than one of them appears in a solution, they are then termed as solitons. If one of these two competing effects is lost, solitons become unstable and eventually cease to exist. In this respect, solitons are completely different from linear waves.

The KdV equation has also been found to describe a number of important physical phenomena such as magnetohydrodynamic waves in a warm plasma [3], elastic waves in an anharmonic crystal [4], ion-acoustic waves in a plasma [5], the long-lived giant red

spot in the highly turbulent Jovian atmosphere and the propagation of short laser pulses in optical fibres [6].

The existence of unique solutions for some classes of initial data has been established in Lax [7]. There exist many integral invariants of the KdV equation, three of these serve as benchmarks to test the efficiency of numerical solvers. If $u(x,t)$ is the solution of the KdV equation given by Equation (1) above, the three invariants are

$$F_1 = \int_0^L u\,dx, \tag{2}$$

$$F_2 = \frac{1}{2}\int_0^L u^2\,dx, \tag{3}$$

$$F_3 = \int_0^L \left(\frac{1}{2}\delta^2(u_x)^2 - \frac{1}{6}u^3\right)dx, \tag{4}$$

which represent mass, momentum and energy conservation, respectively.

Li and vu-Quoc [8] stated that in some areas, the ability to preserve the invariant properties of the original differential equations is a criterion to judge the success of a numerical simulation. The discrete forms of the three conservation laws are

$$F_1^h(\bar{v}) = \sum_{i=1}^n v_i \Delta x, \tag{5}$$

$$F_2^h(\bar{v}) = \frac{1}{2}\sum_{i=1}^n \left(\frac{v_i + v_{i-1}}{2}\right)^2 \Delta x, \tag{6}$$

$$F_3^h(\bar{v}) = \sum_{i=1}^n \left(\frac{1}{2}\delta^2|\Delta_+ v_i|^2 - \frac{1}{6}(v_i)^3\right)\Delta x. \tag{7}$$

There are several good reasons to work with KdV equation as a prototype. The reasons are detailed below:

1. It is a model of a nonlinear hyperbolic equation with smooth solutions for all times.
2. It is non-dissipative and therefore is a natural testbed for comparing conservative vs. dissipative discretization.
3. It is notorious. It is well known that unexpected, nonlinear instabilities occasionally arise from reasonable-looking finite difference method. For instance, Zhao and Qin [9] solved $u_t + \eta u u_x + \delta^2 u_{xxx} = 0$ with $u(t = 0, x) = u_0(x) = \cos(\pi x)$ for $x \in [0,2]$, $\eta = 1$, $\delta = 0.022$ using periodic boundary conditions and employing the Zabusky–Kruskal scheme. They tried various $\Delta x$, $\Delta t$ combinations satisfying the linear stability bound, yet they always obtained blow-up phenomena for $t > \dfrac{21}{\pi}$. Setting $\Delta x = 0.01$, $\Delta t = 0.0001$, the solution, using an explicit scheme, is qualitatively correct for a while. Around $t = 5$, i.e., after 50,000 time steps, error accumulates in the solution so that the linear stability bound is violated. After a few more steps, the solution blows up.

The design and development of symplectic methods for Hamiltonian ODEs has yielded very powerful numerical schemes with beautiful geometric properties. Symplectic and other symmetric methods have been noted for their superior performance, especially for long time integration [10–12]. The Korteweg–de-Vries equation has been extensively studied in numerous studies using symplectic and multisymplectic methods [13]. Because of nonlinear instability, all those methods that can remove the phenomenon of unphysical oscillations are entirely implicit or semi-explicit except for the 6-point scheme that was proposed very recently in [14] based on the concept of multisymplectic schemes. However, due to stability constraint on the time step, the 6-point scheme is somewhat slow. Wang et al. [1] attempted to construct a scheme that not only removes oscillation phenomenon for long time propagation but is also faster than the 6-point scheme.

Very recently, the authors in [15] used two existing schemes proposed by Zabusky and Kruskal [4] and Wang et al. [1] to solve $u_t + 6uu_x + \beta u_{xxx} = 0$ with initial conditions $u(x, t = 0) = 2\mu^2 \text{sech}^2(\mu x)$ with $\mu > 0$ [16]. Appadu et al. [15] also constructed two novel methods obtained by modifying the scheme proposed by Zabusky and Kruskal. The performance of the four methods is compared in regard to dispersion and dissipation errors and ability to conserve mass, momentum and energy by using two numerical experiments that involve solitons. It is worthy to note that sine-cosine and canonical transformation methods proposed and employed in [17,18] have proven useful in obtaining exact soliton solutions and solutions of the Navier–Stokes equation, respectively. We next discuss dispersive characteristics of numerical methods.

The term with the lowest even order spatial derivative in the truncation error produces amplitude error in the numerical solution, and this is responsible for numerical dissipation. The leading odd spatial derivative in the truncation error produces small-scale waves as different Fourier components propagate at different phase speeds, and this causes numerical dispersion [19]. The relative phase error (RPE) is a measure of the dispersive character of a scheme. This quantity is a ratio and measures the velocity of the computed waves to that of the physical waves [20,21]. The relative phase error is obtained using the relation [22]

$$RPE = \frac{arg(\xi_{num})}{arg(\xi_{exact})} \tag{8}$$

where $\xi_{num}$ is the amplification factor of the numerical scheme and $\xi_{exact}$ is the exact amplification factor. The relative phase error of numerical schemes discretizing 1D linear advection equation ($u_t + \beta u_x = 0$) and 1D advection-diffusion equation ($u_t + \beta u_x = \alpha u_{xx}$) is calculated as [21,23–25]

$$RPE = \frac{1}{cw} \arctan\left[\frac{Im(\xi_{num})}{Re(\xi_{num})}\right]. \tag{9}$$

Plots of relative phase error vs. $w$ at some values of $\Delta x$, $\Delta t$ for a few schemes discretizing 1D linear advection and 1D advection-diffusion equation were obtained in [21] and [24], respectively. Plots of RPE vs. $w_x$ vs. $w_y$ were obtained in [23,25] for numerical schemes discretizing 2D advection-diffusion and 3D advection-diffusion equations. Ascher and McLachlan [26] obtained the dispersion relation of the partial differential equation

$$u_t = 2\epsilon u u_x + \rho u_x + \nu u_{xxx} \tag{10}$$

On considering the linearized version of the PDE

$$u_t = \rho u_x + \nu u_{xxx}. \tag{11}$$

Appadu et al. [15] considered the linearized form $u_t + \beta u_{xxx} = 0$ of the PDE $u_t + \gamma u u_x + \beta u_{xxx} = 0$ to study dispersion analysis of the KdV equation. Some work on computing the optimal temporal step size by minimizing the integrated error at a given $\Delta x$ was done by Appadu et al. [27]. The integrated error was obtained as the square of dispersion error.

The paper is organized as follows. The two numerical experiments considered are described in Section 2. Section 3 is devoted to Scheme 1 applied to numerical experiment 1 and details the following: derivation, stability, consistency, presentation of numerical results and dispersion analysis. Sections 4 and 5 give information on work done on Schemes 2 and 3 when used to solve numerical experiment 1. Sections 6–8 concerns the derivation, stability, consistency, presentation of numerical results and dispersion analysis of Schemes 1, 2, 3 when solving numerical experiment 2. We highlight salient features of the paper in Section 9.

## 2. Numerical Experiment

This paper is dedicated to the analysis of three numerical methods for the approximation of two different forms of linear KdV equations.

### 2.1. Case 1 (Numerical Experiment 1)

We consider the linear dispersive KdV equation [28]

$$u_t + 2u_x + u_{xxx} = 0 \tag{12}$$

with $x \in [0, 2\pi]$ and $t \in [0, 4.0]$.
The initial condition is $u(x, 0) = \sin(x)$, boundary conditions are

$$u(0, t) = \sin(-t) \tag{13}$$

and

$$u(2\pi, t) = \sin(2\pi - t). \tag{14}$$

The exact solution is $u(x, t) = \sin(x - t)$. This problem was solved using the Adomian decomposition method [28], Laplace–Adomian decomposition method [29], Homotopy perturbation method [29], Bernstein–Laplace–Adomian method [29] and Reduced Differential Transform method [29].

### 2.2. Case 2 (Numerical Experiment 2)

We considered a new case when the coefficient of dispersion was greater than that of advection. We solve

$$u_t + 2u_x + 5u_{xxx} = 0 \tag{15}$$

with $x \in [0, 2\pi]$ and $t \in [0, 4]$. The initial condition is $u(x, 0) = \sin(x)$, and the boundary conditions are

$$u(0, t) = \sin(3t) \tag{16}$$

and

$$u(2\pi, t) = \sin(2\pi + 3t). \tag{17}$$

The exact solution is $u(x, t) = \sin(x + 3t)$.

## 3. Scheme 1 for Numerical Experiment 1

A classical finite difference scheme given by

$$v_i^{n+1} = v_i^{n-1} - \frac{k}{3h}(v_{i-1}^n + v_i^n + v_{i+1}^n)(v_{i+1}^n - v_{i-1}^n) - \eta^2 \frac{k}{h^3}(v_{i+2}^n - 2v_{i+1}^n + 2v_{i-1}^n - v_{i-2}^n)$$

was proposed in [4] for the solution of the nonlinear equation $u_t + uu_x + \delta^2 u_{xxx} = 0$. The scheme is adapted to solve $u_t + 2u_x + u_{xxx} = 0$ as given in Equation (12). Therefore, Scheme 1 is given by

$$v_i^{n+1} = v_i^{n-1} - \frac{2k}{h}(v_{i+1}^n - v_{i-1}^n) - \frac{k}{h^3}(v_{i+2}^n - 2v_{i+1}^n + 2v_{i-1}^n - v_{i-2}^n), \tag{18}$$

for $i = 2, 3, 4, \cdots, NP - 2$ and $n = 2, 3, \cdots, itmax - 1$.

### 3.1. Stability of Scheme 1 for Numerical Experiment 1

In this section, the stability region of Scheme 1 is obtained. We substitute $v_i^n$ by $\xi^n e^{I\theta ih}$ in Equation (18). This gives

$$\xi = \xi^{-1} - \frac{2k}{h}(e^{I\theta h} - e^{-I\theta h}) - \frac{k}{h^3}(e^{2I\theta h} - 2e^{I\theta h} + 2e^{-I\theta h} - e^{2I\theta h}).$$

After some rearrangements, we get

$$\xi^2 + AI\xi - 1 = 0, \tag{19}$$

where

$$A = 4\sin(w)\left[\frac{k}{h} + \frac{k}{h^3}(\cos(w) - 1)\right]. \tag{20}$$

Therefore,

$$\xi = \frac{1}{2}(-IA \pm \sqrt{-A^2 + 4}). \tag{21}$$

We denote the amplification factor of the physical and the computational modes by $\xi_1$ and $\xi_2$, respectively.

$$\xi_1 = \frac{1}{2}(-IA + \sqrt{4 - A^2}),$$

and

$$\xi_2 = -IA - \sqrt{4 - A^2}.$$

If we seek amplification factor $\xi$ that is not purely imaginary, we require $4 - A^2 \geq 0$, which implies that $|A| \leq 2$. This implies that we need $k$ such that

$$\left|\frac{4k}{h}\sin w + \frac{2k}{h^3}(\sin 2w - 2\sin w)\right| \leq 2.$$

Note that the inequality is dominated by $\frac{2k}{h^3}(\sin(2w) - 2\sin(w))$ as $h \to 0$. The maximum is when $\cos(w) = -\frac{1}{2}$. We obtain the stability region as

$$k \leq \frac{1}{\left|\sqrt{3}\left[\frac{1}{h} - \frac{3}{2}\frac{1}{h^3}\right]\right|}. \tag{22}$$

If we choose $h = \pi/10$, (22) gives $k \leq 0.012775$. We can also obtain the stability region by use of 2D plots, as shown in Figures 1 and 2. The stability region when $h = \pi/10$ is $0 < k \leq 0.012$, as illustrated in Figures 1b and 2b.

The procedure used in this section is the von Neumann approach. This procedure will be employed to determine the expression for the amplification factor of the schemes in the later sections. The region is obtained by imposing $|\xi \leq 1|$.

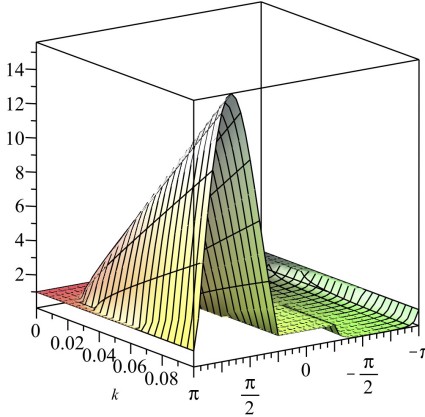
**(a)** Wider range of $k$, $k \in [0, 0.10]$

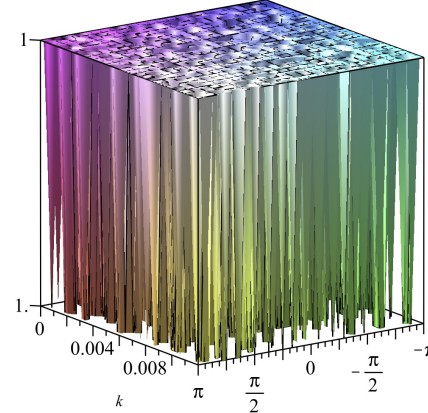
**(b)** Stable region, where $k \in [0, 0.012]$

**Figure 1.** Plot of $|\xi_1|$ vs. $k$ vs. $w \in [-\pi, \pi]$ of Scheme 1 for Numerical Experiment 1. The spatial step size is $h = \pi/10$.

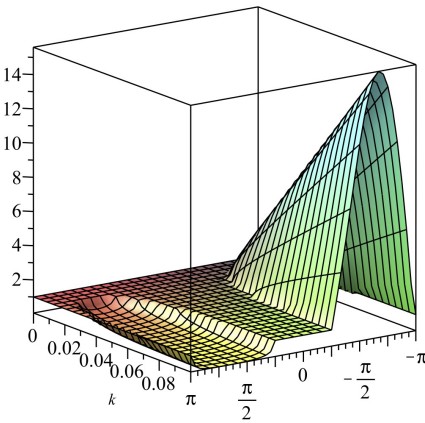
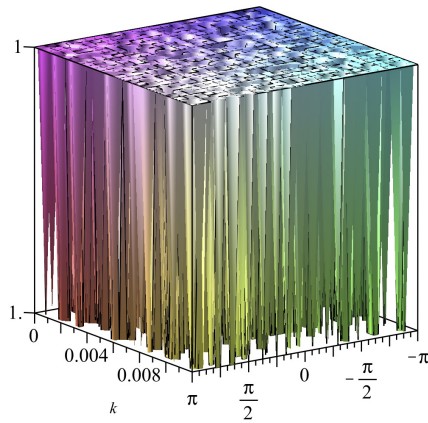

(**a**) Wider range of $k$, $k \in [0, 0.10]$      (**b**) Stable region, where $k \in [0, 0.012]$

**Figure 2.** Plot of $|\xi_2|$ vs. $k$ vs. $w \in [-\pi, \pi]$ of Scheme 1 for Numerical Experiment 1. The spatial step size is $h = \pi/10$.

### 3.2. Consistency of Scheme 1 for Numerical Experiment 1

The scheme in Equation (18) is given by

$$v_i^{n+1} = v_i^{n-1} - \frac{2k}{h}(v_{i+1}^n - v_{i-1}^n) - \frac{k}{h^3}(v_{i+2}^n - 2v_{i+1}^n + 2v_{i-1}^n - v_{i-2}^n).$$

We use the Taylor series about the point $(n, i)$ in order to determine the order of accuracy of the scheme. By the Taylor's expansion,

$$v_i^{n+1} = v + kv_t + \frac{k^2}{2}v_{tt} + \frac{k^3}{6}v_{ttt} + \frac{k^4}{24}v_{tttt} + \dots$$

$$v_i^{n-1} = v - kv_t + \frac{k^2}{2}v_{tt} - \frac{k^3}{6}v_{ttt} + \frac{k^4}{24}v_{tttt} + \dots$$

so that

$$v_i^{n+1} - v_i^{n-1} = 2kv_t + \frac{2k^3}{6}v_{ttt} + \dots$$

Furthermore

$$v_{i+1}^n = v + hv_x + \frac{h^2}{2}v_{xx} + \frac{h^3}{6}v_{xxx} + \frac{h^4}{24}v_{xxxx} + \dots$$

and

$$v_{i-1}^n = v - hv_x + \frac{h^2}{2}v_{xx} - \frac{h^3}{6}v_{xxx} + \frac{h^4}{24}v_{xxxx} + \dots$$

so that

$$v_{i+1}^n - v_{i-1}^n = 2hv_x + \frac{2h^3}{6}v_{xxx} + \dots$$

In addition

$$v_{i+2}^n = v + 2hv_x + \frac{(2h)^2}{2}v_{xx} + \frac{(2h)^3}{6}v_{xxx} + \frac{(2h)^4}{24}v_{xxxx} + \dots$$

and

$$v_{i-2}^n = v - 2hv_x + \frac{(2h)^2}{2}v_{xx} - \frac{(2h)^3}{6}v_{xxx} + \frac{(2h)^4}{24}v_{xxxx} + \dots$$

so that

$$v_{i+2}^n - v_{i-2}^n = 4hv_x + \frac{2(2h)^3}{6}v_{xxx} + \dots$$

We obtain

$$v_t + 2v_x + v_{xxx} = -\frac{k^2}{6}v_{ttt} - \frac{1}{3}h^2 v_{xxx} + \dots$$

The accuracy of the scheme is quadratic both in space and in time.

The detailed consistency analysis is discussed above. The same procedure will be followed throughout the entire paper. We may not discuss it in detail for other methods.

### 3.3. Numerical Results

In this section, the approximation of the linear dispersive Equation (12) using Scheme 1 given in Equation (18) is presented. Within the stability region, the spatial and the temporal step sizes employed are $h = \pi/10$ and $k = 0.001$, respectively. The solution profiles are shown in Figure 3. These are compared with the exact solution $u(x,t) = \sin(x - t)$. The profile of absolute errors vs. $x$ is also presented at $T = 2.0$ and $T = 4.0$. Figure 4 displays the absolute error profiles for the scheme. Table 1 displays $L_1$ and $L_\infty$ errors at some values of $k$ when $h = \pi/10$.

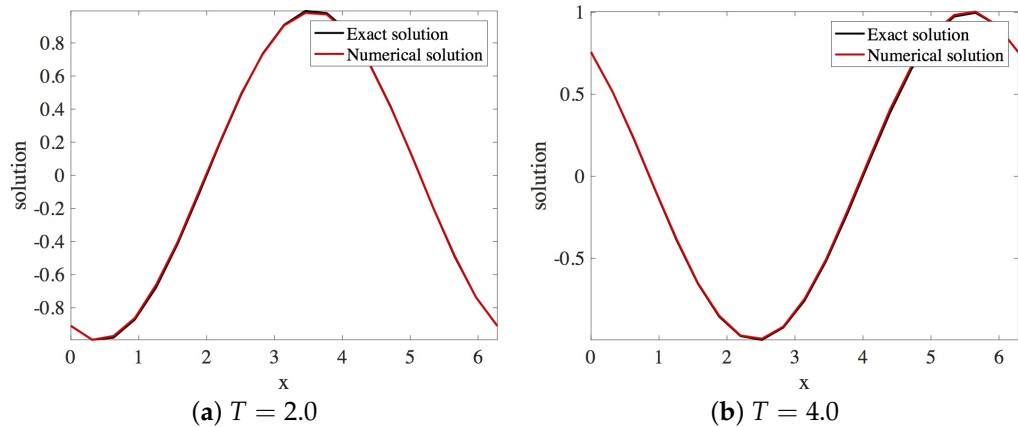

(**a**) $T = 2.0$        (**b**) $T = 4.0$

**Figure 3.** Plot of exact and numerical profiles (using Scheme 1), vs. $x$ at times 2.0 and 4.0 using $h = \pi/10$ and $k = 0.001$ (Numerical Experiment 1).

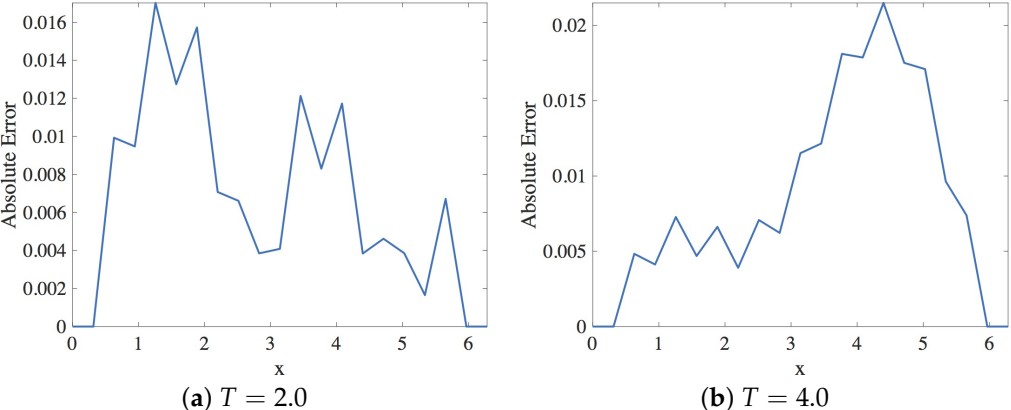

(**a**) $T = 2.0$        (**b**) $T = 4.0$

**Figure 4.** Plot of the absolute error vs. $x$ using Scheme 1 for Numerical Experiment 1. The spatial and temporal step sizes are $h = \pi/10$ and $k = 0.001$, respectively

**Table 1.** $L_1$ and $L_\infty$ errors for various $k$ when $h = \pi/10$ when Scheme 1 is employed to approximate Numerical Experiment 1 at $T = 2$ and $T = 4$.

| Step Sizes $k$ | Error | | | |
|:---:|:---:|:---:|:---:|:---:|
| | $T = 2$ | | $T = 4$ | |
| | $L_1$ ($\times 10^{-1}$) | $L_\infty$ ($\times 10^{-2}$) | $L_1$ ($\times 10^{-1}$) | $L_\infty$ ($\times 10^{-2}$) |
| 0.001 | 1.3933 | 1.7015 | 1.7752 | 2.1496 |
| 0.002 | 1.3933 | 1.7016 | 1.7751 | 2.1493 |
| 0.003 | 1.3936 | 1.7021 | 1.7753 | 2.1477 |
| 0.004 | 1.3934 | 1.7019 | 1.7747 | 2.1478 |
| 0.005 | 1.3934 | 1.7012 | 1.7746 | 2.1464 |
| 0.006 | 1.3918 | 1.6964 | 1.7736 | 2.1510 |
| 0.007 | 1.3943 | 1.6997 | 1.7752 | 2.1442 |
| 0.008 | 1.3936 | 1.6998 | 1.7738 | 2.1418 |
| 0.009 | 1.3934 | 1.7011 | 1.7752 | 2.1362 |
| 0.01 | 1.3944 | 1.7012 | 1.7725 | 2.1368 |

*3.4. Dispersion Analysis*

A perturbation for $u(x,t)$ is

$$e^{\alpha t + I\theta x}, \tag{23}$$

where $\alpha$ is the dispersion relation. We consider $u_t + 2u_x + u_{xxx} = 0$. Then, $u_t = \alpha u(x,t)$, $u_x = I\theta u(x,t)$, $u_{xxx} = -I\theta^3 u(x,t)$. Substituting these values in Equation (12), we obtain

$$\alpha + 2I\theta - I\theta^3 = 0,$$

leading to

$$\alpha = I\theta(\theta^2 - 2).$$

This implies that the perturbation for $u(x,t)$ can be written as a function of $\theta$ alone as $e^{I\theta[(\theta^2-2)t+x]}$. The amplification factor is determined from the relation

$$\xi_{exact} = \frac{u(x,t^{n+1})}{u(x,t^n)} = e^{I\theta(\theta^2-2)k}.$$

Relative phase error (RPE) is defined [27] as

$$RPE = \frac{arg(\xi_{numerical})}{arg(\xi_{exact})}.$$

We obtain plots of the arguments of $\xi_{exact}$, $\xi_1$, $\xi_2$ vs. $w \in [0, \pi]$ in Figure 5. For values of $w \in [0, 1]$, the graphs of the arguments of both $\xi_{exact}$ and $\xi_1$ are very close to each other, as illustrated in Figure 5a. For values of $w \in [0, 0.5]$, the graphs for the arguments of $\xi_{exact}$ and $\xi_2$ are very close to each other, as depicted in Figure 5b, and we note that $arg(\xi_2)$ and $arg(\xi_{exact})$ are of opposite signs for $0.5 < w < \pi$.

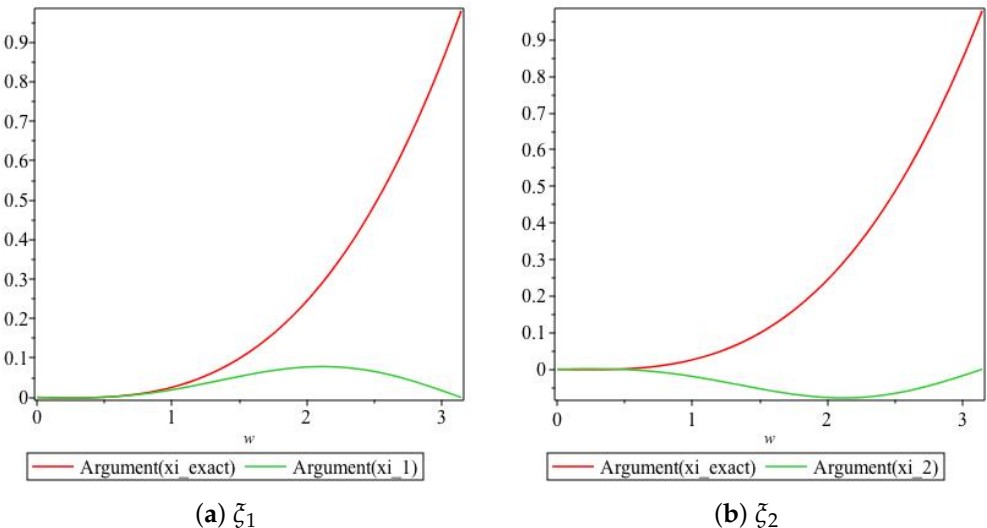

**Figure 5.** Plot of the arguments of the exact and numerical amplification factors of Scheme 1 for Numerical Experiment 1 with the spatial and temporal step sizes $h = \pi/10$ and $k = 0.001$, respectively.

## 4. Scheme 2 for Numerical Experiment 1

In 2007, Wang et al. [14] constructed the following scheme to solve

$$u_t + \eta u u_x + \delta^2 u_{xxx} = 0 : \tag{24}$$

$$\frac{v_i^{n+1} - v_i^n + v_{i+1}^n - v_{i+1}^{n-1}}{2k} + \frac{\eta}{2h}(v_{i+1}^n + v_i^n)(v_{i+1}^n - v_i^n) + \frac{\mu^2}{h^3}(v_{i+2}^n - 3v_{i+1}^n + 3v_i^n - v_{i-1}^n) = 0.$$

where $\eta$ is the coefficient of the nonlinear term $uu_x$. The scheme is multisymplectic and can remove the dispersive oscillations and preserves approximately several conservation laws of the KdV equation [14].

To discretize $u_t + 2u_x + u_{xxx} = 0$, we propose

$$\frac{1}{2k}(v_i^{n+1} - v_i^n + v_{i+1}^n - v_{i+1}^{n-1}) + \frac{2}{h}(v_{i+1}^n - v_i^n) + \frac{1}{h^3}(v_{i+2}^n - 3v_{i+1}^n + 3v_i^n - v_{i-1}^n) = 0.$$

This gives

$$v_i^{n+1} - v_i^n + v_{i+1}^n - v_{i+1}^{n-1} = -\frac{4k}{h}(v_{i+1}^n - v_i^n) - \frac{2k}{h^3}(v_{i+2}^n - 3v_{i+1}^n + 3v_i^n - v_{i-1}^n),$$

which can be rewritten as

$$v_i^{n+1} = v_i^n - v_{i+1}^n + v_{i+1}^{n-1} - \frac{4k}{h}(v_{i+1}^n - v_i^n) - \frac{2k}{h^3}(v_{i+2}^n - 3v_{i+1}^n + 3v_i^n - v_{i-1}^n), \tag{25}$$

for $i = 2, 3, \cdots, NP - 2$ and $n = 2, 3, \cdots, itmax - 1$.

### 4.1. Stability of Scheme 2 for Numerical Experiment 1

Substituting $v_i^n$ by $\xi^n e^{I\theta ih}$ and simplifying gives

$$\xi = 1 - e^{Iw} + \xi^{-1} e^{Iw} - \frac{4k}{h}(e^{Iw} - 1) - \frac{2k}{h^3}(e^{2Iw} - 3e^{Iw} + 3 - e^{-Iw}).$$

After some rearrangements, we get where

$$A = -(1 - \cos(w) - I\sin(w)) + \frac{4k}{h}(\cos(w) - 1 + I\sin(w))$$

$$+\frac{2k}{h^3}(\cos(2w) + I\sin(2w) - 3(\cos(w) + I\sin(w)) + 3 - (\cos(w) - I\sin(w)))$$

$$B = -(\cos(w) + I\sin(w))$$

and therefore, we get

$$\xi_{1,2} = \frac{1}{2}(-A \pm \sqrt{A^2 - 4B}),$$

with $\xi_1 = \frac{1}{2}(-A + \sqrt{A^2 - 4B})$ and $\xi_2 = \frac{1}{2}(-A - \sqrt{A^2 - 4B})$. The range of values of the temporal step size $k$ for which $|\xi| \leq 1$ is shown in the profiles of the amplification factor $\xi$ (see Figures 6 and 7) given that $w = \theta h \in [-\pi, \pi]$ and $h = \frac{\pi}{10}$. It is observed that $0 < k < 0.01$.

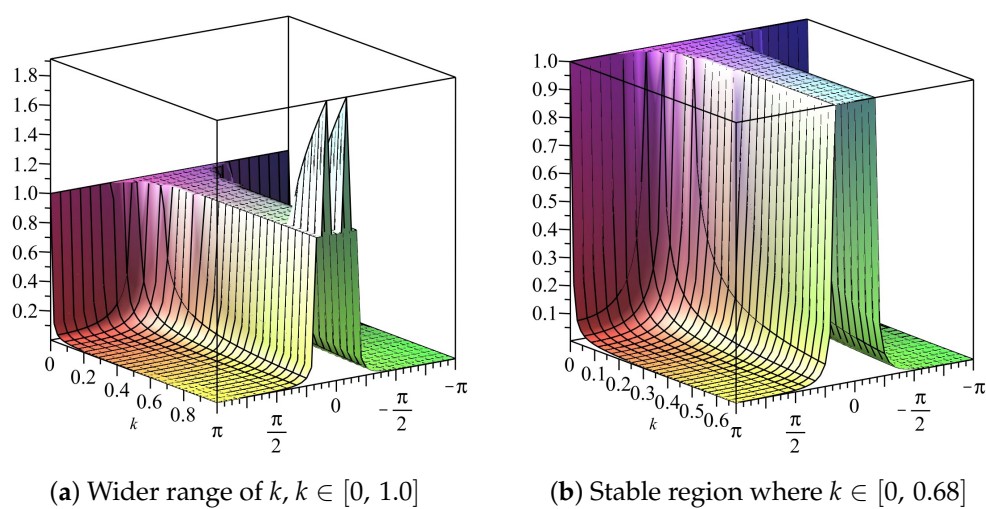

(**a**) Wider range of $k$, $k \in [0, 1.0]$      (**b**) Stable region where $k \in [0, 0.68]$

**Figure 6.** Plot of $|\xi_1|$ vs. $k$ vs. $w \in [-\pi, \pi]$ of Scheme 2 for Numerical Experiment 1. The spatial step size $h = \pi/10$.

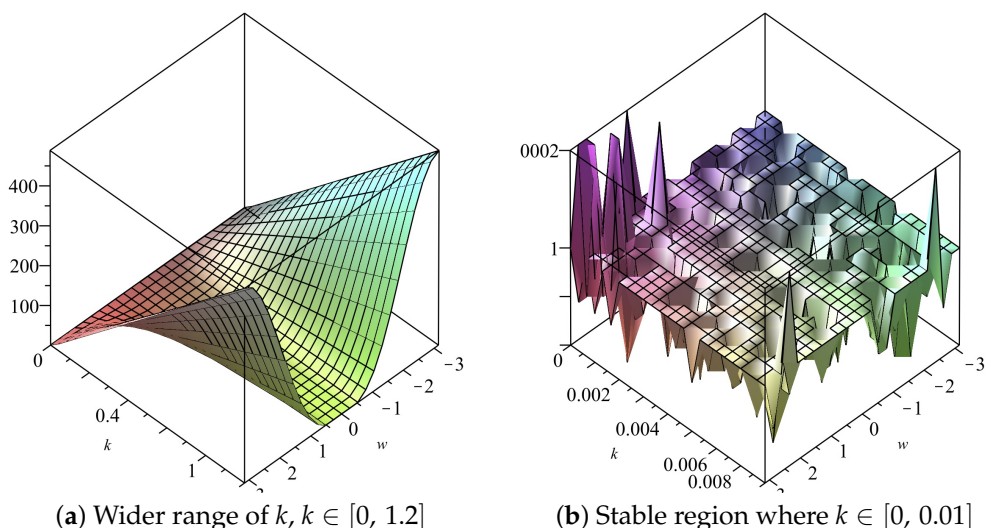

(**a**) Wider range of $k$, $k \in [0, 1.2]$      (**b**) Stable region where $k \in [0, 0.01]$

**Figure 7.** Plot of $|\xi_2|$ vs. $k$ vs. $w \in [-\pi, \pi]$ of Scheme 2 for Numerical Experiment 1. The spatial step size $h = \pi/10$.

### 4.2. Consistency of Scheme 2 for Numerical Experiment 1

We recall that the scheme is given by

$$v_i^{n+1} = v_i^n - v_{i+1}^n + v_{i+1}^{n-1} - \frac{4k}{h}(v_{i+1}^n - v_i^n) - \frac{2k}{h^3}(v_{i+2}^n - 3v_{i+1}^n + 3v_i^n - v_{i-1}^n).$$

Taylor series expansion about $(n, i)$ after simplification gives

$$2kv_t + \frac{1}{3}k^3 v_{ttt} + khv_{tx} - \frac{1}{2}k^2 hv_{ttx} + \frac{1}{2}kh^2 v_{txx} + 4kv_x + 2khv_{xx} + \frac{2}{3}kh^2 v_{xxx} + 2kv_{xxx} + \ldots = 0.$$

We obtain

$$v_t + 2v_x + v_{xxx} = -\frac{1}{6}k^2 v_{ttt} - \frac{h}{2}v_{tx} + \frac{kh}{4}v_{ttx} - \frac{h^2}{4}v_{txx} - hv_{xx} + \cdots$$

Since $v_{tx} + 2v_{xx} + v_{xxxx} \approx 0$, hence

$$v_t + 2v_x + v_{xxx} = -\frac{1}{6}k^2 v_{ttt} + \frac{kh}{4}v_{ttx} - \frac{h^2}{4}v_{txx} - \frac{1}{3}h^2 v_{xxx} + \cdots.$$

The last equation shows that the scheme is second order accurate both spatially and temporally.

### 4.3. Numerical Results

In this section, the approximation of the linear dispersive Equation (12) by Scheme 2 given in (25) is presented. Within the stability region, the spatial and the temporal step size employed are $h = \pi/10$ and $k = 0.0001$, respectively. The solution profiles are shown in Figure 8. These are compared with exact solution $u(x,t) = \sin(x - t)$. The profile of absolute errors vs. $x$ is also presented at $T = 2.0$ and $T = 4.0$. Figure 9 displays the absolute error profiles for the multisymplectic scheme. We display $L_1$ and $L_\infty$ errors in Table 2.

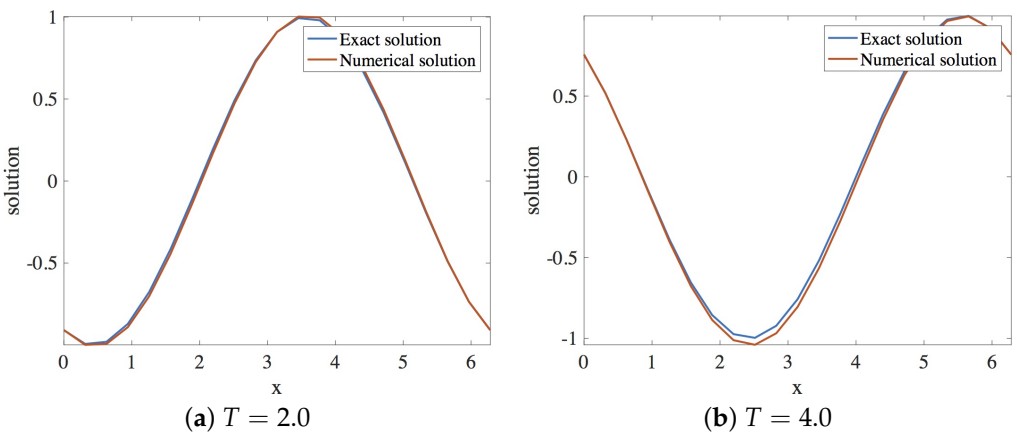

(**a**) $T = 2.0$        (**b**) $T = 4.0$

**Figure 8.** Plot of exact and numerical profiles (using Scheme 2), vs. $x$ at times 2.0 and 4.0 using $h = \pi/10$ and $k = 0.001$.

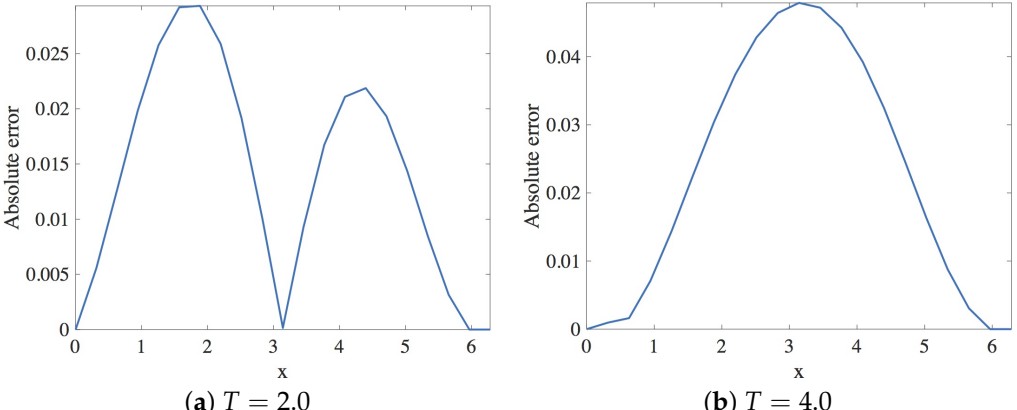

(**a**) $T = 2.0$        (**b**) $T = 4.0$

**Figure 9.** Plot of the absolute error vs. $x$ using Scheme 2 for Numerical Experiment 1. The spatial and temporal step sizes are $h = \pi/10$ and $k = 0.001$, respectively.

**Table 2.** $L_1$ and $L_\infty$ errors for various $k$ when $h = \pi/10$ when Scheme 2 is employed to approximate Numerical Experiment 1 at $T = 2$ and $T = 4$.

| Step Sizes $k$ | Error | | | |
|---|---|---|---|---|
| | $T = 2$ | | $T = 4$ | |
| | $L_1$ ($\times 10^{-1}$) | $L_\infty$ ($\times 10^{-2}$) | $L_1$ ($\times 10^{-1}$) | $L_\infty$ ($\times 10^{-2}$) |
| 0.001 | 2.9200 | 2.9327 | 4.6733 | 4.7894 |
| 0.002 | 2.9335 | 2.9464 | 4.6955 | 4.8124 |
| 0.003 | 2.9486 | 2.9628 | 4.7174 | 4.8344 |
| 0.004 | 2.9607 | 2.9739 | 4.7402 | 4.8587 |
| 0.005 | 2.9743 | 2.9877 | 4.7627 | 4.8819 |
| 0.006 | 2.9849 | 2.9961 | 4.7860 | 4.9074 |
| 0.007 | 3.0047 | 3.0219 | 4.8069 | 4.9244 |
| 0.008 | 3.0122 | 3.0345 | 4.8314 | 4.9580 |

*4.4. Dispersion Analysis*

Arguments of numerical amplification factors $\xi_{1,2}$ derived in Section 4.1 are compared with the exact amplification factor within the stability region. The comparison is shown in Figure 10. We obtain plots of the arguments of $\xi_{exact}$, $\xi_1$, $\xi_2$ vs. $w \in [0, \pi]$ in Figure 10. For values of $w \in [0, 1]$, the graphs for the arguments of both $\xi_{exact}$ and $\xi_1$ are very close to each other, as illustrated in Figure 10a. Figure 10b shows that the arguments of $\xi_{exact}$ and $\xi_1$ are not close for even small values of $w$.

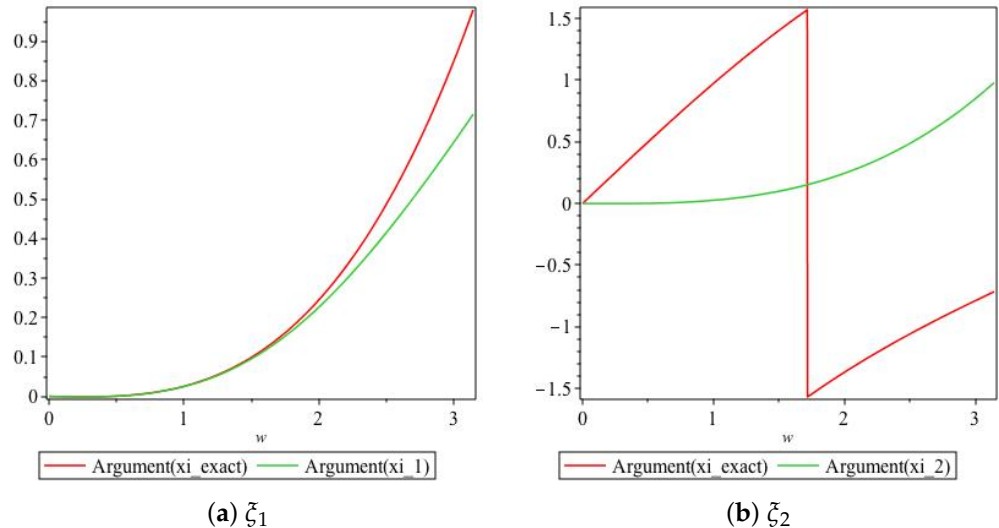

(**a**) $\xi_1$        (**b**) $\xi_2$

**Figure 10.** Plot of the arguments of amplification factors of Scheme 2 for Numerical Experiment 1 with spatial and temporal step sizes $h = \pi/10$ and $k = 0.001$, respectively

## 5. Scheme 3 for Numerical Experiment 1

In 2008, Wang et al. [1] proposed the following scheme to discretize $u_t + \eta u u_x + \delta^2 u_{xxx} = 0$, namely:

$$\frac{1}{2k}(v_{i-1}^{n+1} - v_{i-1}^n + v_{i+1}^n - v_{i+1}^{n-1}) + \eta \left( \frac{v_{i-1}^n + v_i^n + v_{i+1}^n}{3} \right) \left( \frac{v_{i+1}^n - v_{i-1}^n}{2h} \right)$$

$$+ \frac{\mu^2}{2h^3}(v_{i+2}^n - 2v_{i+1}^n + 2v_{i-1}^n - v_{i-2}^n) = 0.$$

In order to discretize $u_t + 2u_x + u_{xxx} = 0$, we propose

$$\frac{1}{2k}(v_{i-1}^{n+1} - v_{i-1}^{n} + v_{i+1}^{n} - v_{i+1}^{n-1}) + \frac{1}{h}(v_{i+1}^{n} - v_{i-1}^{n}) + \frac{1}{2h^3}(v_{i+2}^{n} - 2v_{i+1}^{n} + 2v_{i-1}^{n} - v_{i-2}^{n}) = 0.$$

This gives

$$v_{i-1}^{n+1} - v_{i-1}^{n} + v_{i+1}^{n} - v_{i+1}^{n-1} = -\frac{2k}{h}(v_{i+1}^{n} - v_{i-1}^{n}) - \frac{k}{h^3}(v_{i+2}^{n} - 2v_{i+1}^{n} + 2v_{i-1}^{n} - v_{i-2}^{n}) = 0,$$

which can be written explicitly as

$$v_i^{n+1} = v_i^n - v_{i+2}^n + v_{i+2}^{n-1} - \frac{2k}{h}(v_{i+2}^n - v_i^n) - \frac{k}{h^3}(v_{i+3}^n - 2v_{i+2}^n + 2v_i^n - v_{i-1}^n), \tag{26}$$

where $i = 2, 3, \ldots, NP - 3$ and $n = 2, 3, \ldots, itmax - 1$.

### 5.1. Stability of Scheme 3 for Numerical Experiment 1

Substituting $v_i^n$ by $\xi^n e^{I\theta ih}$ and simplifying gives

$$\xi = 1 - e^{2I\theta h} + \xi^{-1}e^{2I\theta h} - \frac{2k}{h}(e^{2I\theta h} - 1) - \frac{k}{h^3}(e^{3I\theta h} - 2e^{2I\theta h} + 2 - e^{-I\theta h}),$$

After some rearrangements, we get

$$\xi^2 + A\xi + B = 0, \tag{27}$$

where

$$A = \frac{2k}{h}(\cos(2w) - 1 + I\sin(2w)) +$$

$$\frac{k}{h^3}(\cos(3w) - 2\cos(2w) + 2 - \cos(w) + I[\sin(3w) - 2\sin(2w) + \sin(w)])$$

$$-(1 - \cos(2w) - I\sin(2w))$$

and

$$B = -(\cos(2w) + I\sin(2w)).$$

Therefore,

$$\xi_{1,2} = \frac{1}{2}(-A \pm \sqrt{A^2 - 4B}). \tag{28}$$

We employ the condition $|\xi| \leq 1$ to determine the appropriate range for the temporal step size when the spatial step size is $\frac{\pi}{10}$ with $w \in [-\pi, \pi]$. The temporal step size is based on the behaviour of both $\xi_1 = \frac{1}{2}(-A + \sqrt{A^2 - 4B})$ and $\xi_2 = \frac{1}{2}(-A - \sqrt{A^2 - 4B})$. The region of stability, as shown in Figures 11 and 12, implies that the scheme is stable for all $0 < k \leq 0.001$.

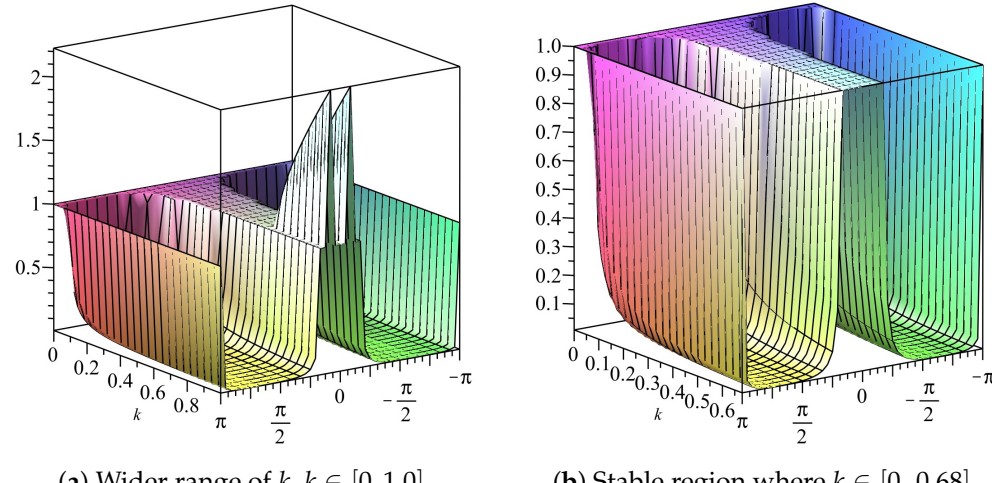

(**a**) Wider range of $k$, $k \in [0, 1.0]$       (**b**) Stable region where $k \in [0, 0.68]$

**Figure 11.** Plot of $|\xi_1|$ vs. $k$ vs. $w \in [-\pi, \pi]$ of Scheme 3 for Numerical Experiment 1. The spatial step size $h = \pi/10$.

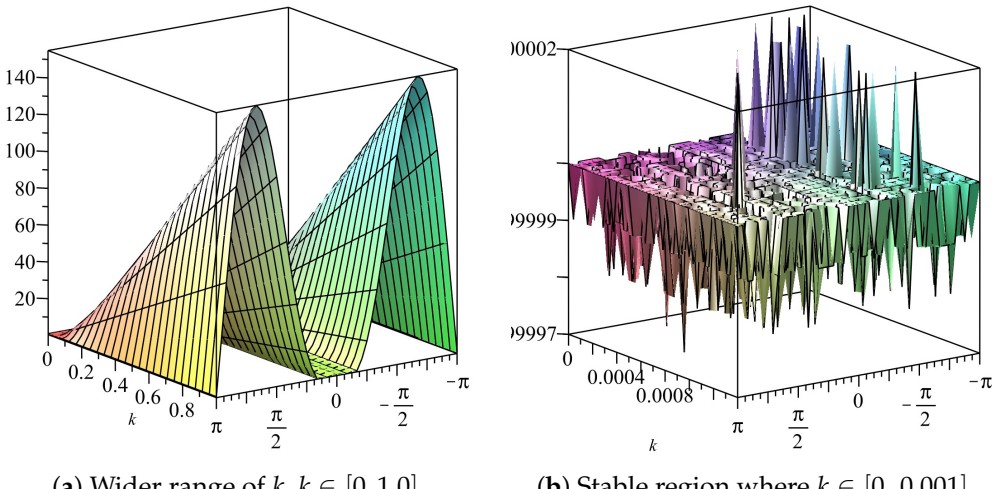

(**a**) Wider range of $k$, $k \in [0, 1.0]$       (**b**) Stable region where $k \in [0, 0.001]$

**Figure 12.** Plot of $|\xi_2|$ vs. $k$ vs. $w \in [-\pi, \pi]$ of Scheme 3 for Numerical Experiment 1. The spatial step size $h = \pi/10$.

*5.2. Consistency of Scheme 3 for Numerical Experiment 1*

In this section, the pointwise accuracy of Scheme (26) will be discussed. The scheme is written as

$$v_i^{n+1} = v_i^n - v_{i+2}^n + v_{i+2}^{n-1} - \frac{2k}{h}(v_{i+2}^n - v_i^n) - \frac{k}{h^3}(v_{i+3}^n - 2v_{i+2}^n + 2v_i^n - v_{i-1}^n).$$

Using Taylor's series expansion about $(n, i)$ gives

$$v + kv_t + \frac{k^2}{2}v_{tt} + \frac{k^3}{6}v_{ttt} + \ldots = v - (v + 2hv_x + \frac{(2h)^2}{2}v_{xx} + \frac{(2h)^3}{6}v_{xxx} + \ldots) + v - kv_t + 2hv_x$$

$$+ \frac{1}{2}[k^2v_{tt} - 4khv_{tx} + 4h^2v_{xx}]$$

$$+ \frac{1}{6}[-k^3v_{ttt} + 6k^2hv_{ttx} - 12kh^2v_{txx} + 8h^3v_{xxx}] - \frac{2k}{h}[v + 2hv_x + \frac{(2h)^2}{2}v_{xx} + \frac{(2h)^3}{6} - v] -$$

$$\frac{k}{h^3}[v + 3hv_x + \frac{(3h)^2}{2}v_{xx} + \frac{(3h)^3}{6}v_{xxx}]$$

$$-\frac{k}{h^3}\left[-2\left(v+2hv_x+\frac{(2h)^2}{2}v_{xx}+\frac{(2h)^3}{6}v_{xxx}\right)+2v-\left(v-hv_x+\frac{h^2}{2}v_{xx}-\frac{h^3}{6}v_{xxx}\right)\right].$$

After some simplifications, we get,

$$v_t+2v_x+v_{xxx}=-\frac{1}{6}k^2v_{ttt}+hv_{xxxx}-\frac{kh}{2}v_{ttx}-h^2v_{txx}-\frac{4}{3}h^2v_{xxx}+\dots.$$

We conclude here that the scheme is second order accurate both in space and time.

### 5.3. Numerical Results

In this section, the approximation of the linear dispersive Equation (12) using Scheme 3 is presented. Within the stability region, the spatial and the temporal step size employed are $h=\pi/10$ and $k=0.001$, respectively. The solution profiles are shown in Figure 13. These are compared with exact solution $u(x,t)=\sin(x-t)$. The profile of absolute errors is also presented at $T=2.0$ and $T=4.0$. Figure 14 displays the absolute error profiles for the multisymplectic scheme. We compare $L_1$ and $L_\infty$ errors in Table 3.

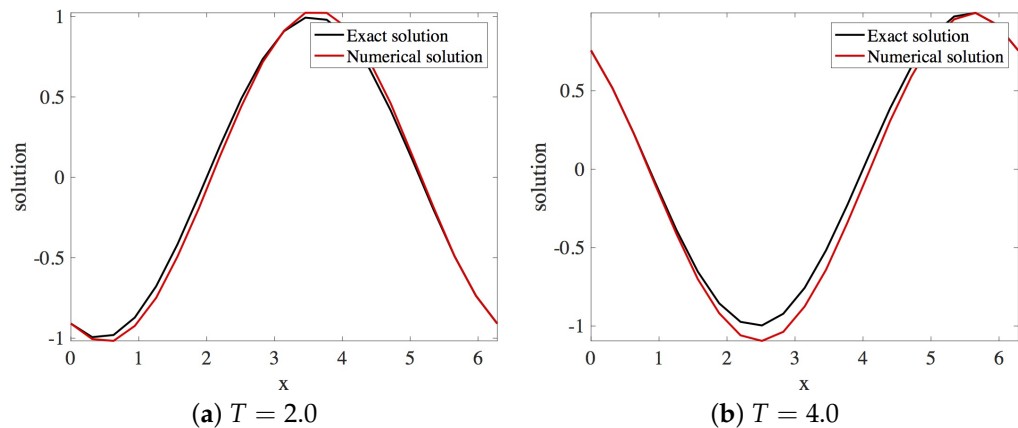

(**a**) $T=2.0$      (**b**) $T=4.0$

**Figure 13.** Plot of the solution profile of Numerical Experiment 1 by Scheme 3. The spatial and temporal step sizes are $h=\pi/10$ and $k=0.001$, respectively.

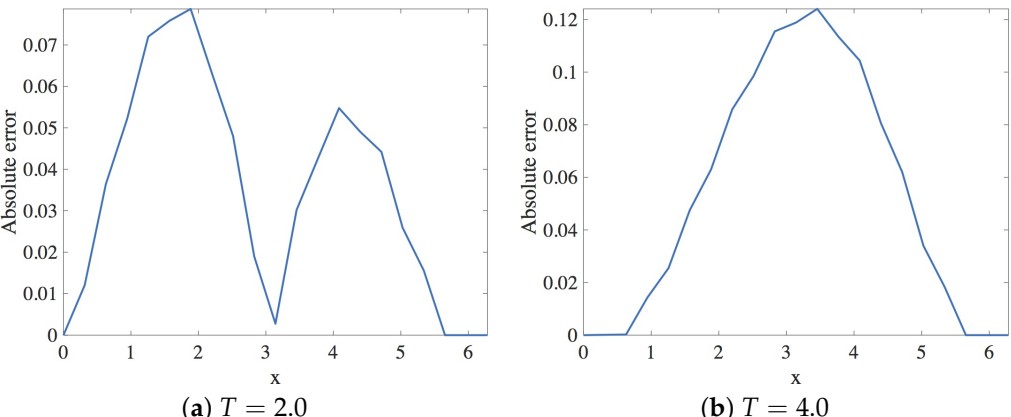

(**a**) $T=2.0$      (**b**) $T=4.0$

**Figure 14.** Plot of the absolute error vs. $x$ using Scheme 3 for Numerical Experiment 1. The spatial and temporal step sizes are $h=\pi/10$ and $k=0.001$, respectively.

**Table 3.** $L_1$ and $L_\infty$ errors for various $k$ when $h = \pi/10$ when Scheme 3 is employed to approximate Numerical Experiment 1 at $T = 2$ and $T = 4$.

| Step Sizes $k$ | Error | | | |
|---|---|---|---|---|
| | $T = 2$ | | $T = 4$ | |
| | $L_1$ ($\times 10^{-1}$) | $L_\infty$ ($\times 10^{-2}$) | $L_1$ | $L_\infty$ ($\times 10^{-1}$) |
| 0.001 | 7.2243 | 7.8696 | 1.1062 | 1.2406 |
| 0.002 | 7.2513 | 7.8993 | 1.1105 | 1.2455 |
| 0.003 | 7.2807 | 7.9356 | 1.1148 | 1.2497 |
| 0.004 | 7.3052 | 7.9589 | 1.1190 | 1.2553 |
| 0.005 | 7.3322 | 7.9887 | 1.1234 | 1.2602 |
| 0.006 | 7.3542 | 8.0054 | 1.1277 | 1.2665 |
| 0.007 | 7.3916 | 8.0616 | 1.1321 | 1.2679 |
| 0.008 | 7.4137 | 8.0783 | 1.1364 | 1.2750 |
| 0.009 | 7.4357 | 8.0950 | 1.1408 | 1.2770 |
| 0.01 | 7.4682 | 8.1382 | 1.1451 | 1.2848 |

*5.4. Dispersion Analysis*

The arguments of the numerical amplification factors $\xi_{1,2}$ derived in Section 7.1 are compared with the exact amplification factor within the stability region. The profiles of the arguments of the amplification factors are shown in Figure 15.

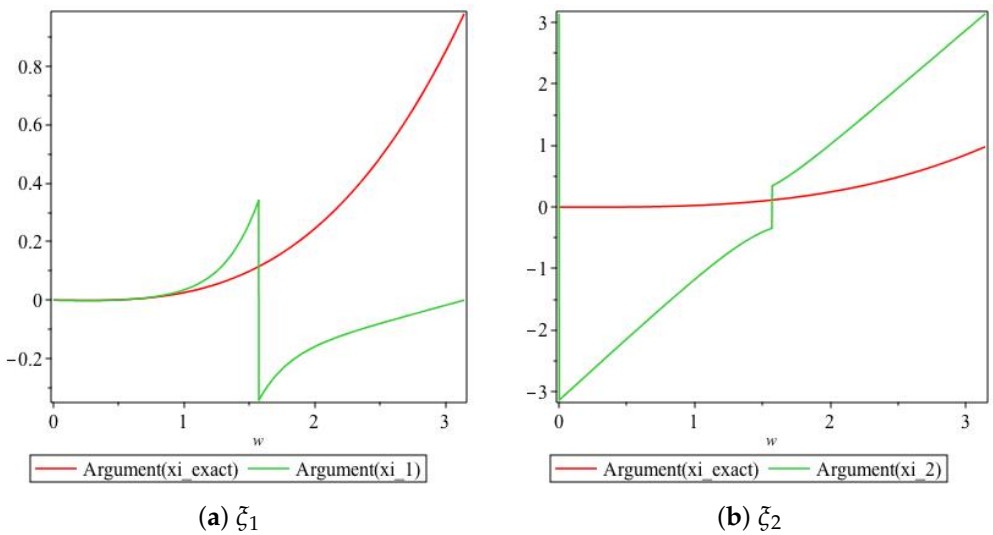

(**a**) $\xi_1$      (**b**) $\xi_2$

**Figure 15.** Plot of the arguments of amplification factors of Scheme 3 for Numerical Experiment 1 with the spatial and temporal step sizes $h = \pi/10$ and $k = 0.001$, respectively.

## 6. Scheme 1 for Numerical Experiment 2

A classical finite difference scheme given by

$$v_i^{n+1} = v_i^{n-1} - \frac{k}{3h}(v_{i-1}^n + v_i^n + v_{i+1}^n)(v_{i+1}^n - v_{i-1}^n) + \eta^2 \frac{k}{h^3}(v_{i+2}^n - 2v_{i+1}^n + 2v_{i-1}^n - v_{i-2}^n)$$

was proposed by Zabusky and Kruskal [4] for the solution of the nonlinear equation $u_t + uu_x + \delta^2 u_{xxx} = 0$. The scheme will be adapted to solve $u_t + 2u_x + 5u_{xxx} = 0$ and will be referred to as Scheme 1. Scheme 1 is given by

$$v_i^{n+1} = v_i^{n-1} - \frac{2k}{h}(v_{i+1}^n - v_{i-1}^n) + \frac{5k}{h^3}(v_{i+2}^n - 2v_{i+1}^n + 2v_{i-1}^n - v_{i-2}^n), \qquad (29)$$

where $i = 1, 2, 3, \ldots, NP - 3$ and $n = 2, 3, \ldots, itmax - 1$.

### 6.1. Stability of Scheme 1 for Numerical Experiment 2

The amplification factor of this scheme satisfies the equation

$$\xi^2 + AI\xi - 1 = 0,$$

where

$$A = 4\sin(w)\left[\frac{k}{h} + \frac{5k}{h^3}(\cos(w) - 1)\right].$$

Therefore,

$$\xi_{1,2} = \frac{1}{2}(-AI \pm \sqrt{-A^2 + 4}).$$

If we seek the amplification factor that is not purely imaginary, we require $4 - A^2 \geq 0$ which implies that $|A| \leq 2$. This implies that we need $k$ such that

$$\left|4\sin(w)\left[\frac{k}{h} + \frac{5k}{h^3}(\cos(w) - 1)\right]\right| \leq 2.$$

Note that the inequality is dominated by $\dfrac{10k}{h^3}(\sin(2w) - 2\sin(w))$ as $h \to 0$. The maximum of this is at the point $\cos(w) = -\dfrac{1}{2}$. The stability region is

$$k \leq \frac{1}{\left|\sqrt{3}\left[\frac{1}{h} - \frac{3}{2}\frac{5}{h^3}\right]\right|}.$$

If we choose $h = \pi/10$, we obtain $k \leq 0.00241869$. The range of values of the temporal step size $k$ for which $|\xi| \leq 1$ is also shown in the profile of the amplification factors $\xi$ shown in Figures 16 and 17 given that $w = \theta h \in [-\pi, \pi]$ and $h = \dfrac{\pi}{10}$. The profile is shown both for $\xi_1 = \dfrac{1}{2}(-IA + \sqrt{-A^2 + 4})$ as well as $\xi_1 = \dfrac{1}{2}(-IA - \sqrt{-A^2 + 4})$. It can be deduced from the two profiles in Figures 16 and 17 that $0 < k \leq 0.0025$.

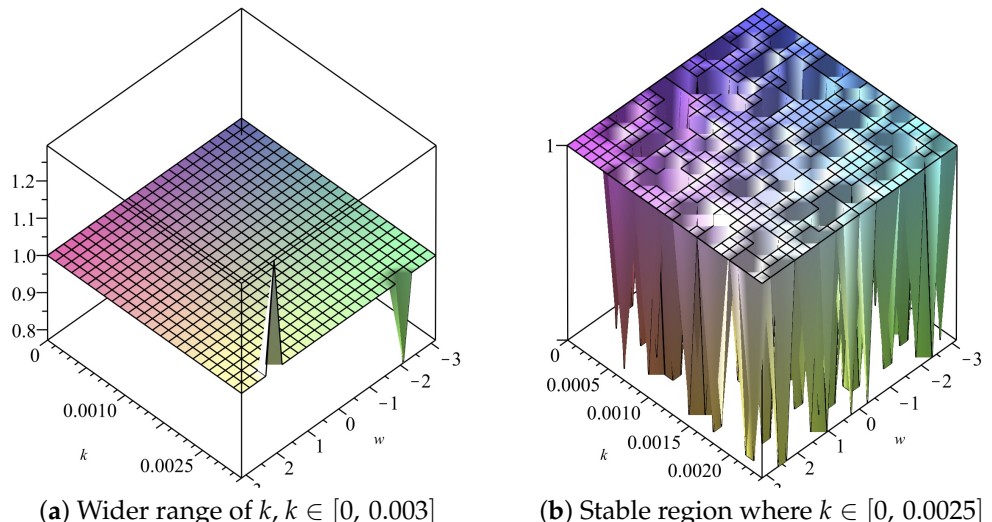

(a) Wider range of $k$, $k \in [0, 0.003]$      (b) Stable region where $k \in [0, 0.0025]$

**Figure 16.** Plot of $|\xi_1|$ vs. $k$ vs. $w \in [-\pi, \pi]$ of Scheme 1 for Numerical Experiment 2. The spatial step size $h = \pi/10$.

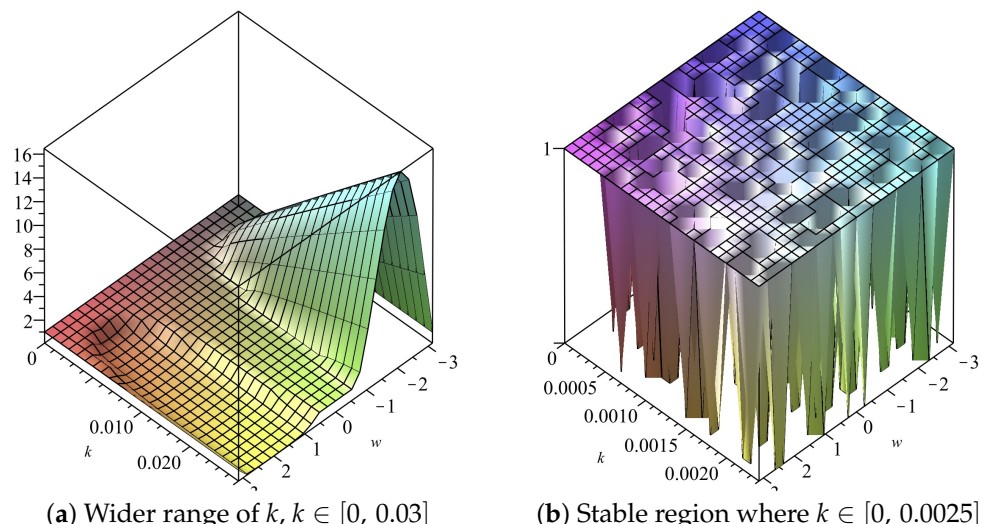

(a) Wider range of $k$, $k \in [0, 0.03]$      (b) Stable region where $k \in [0, 0.0025]$

**Figure 17.** Plot of $|\xi_2|$ vs. $k$ vs. $w \in [-\pi, \pi]$ of Scheme 1 for Numerical Experiment 2. The spatial step size $h = \pi/10$.

### 6.2. Consistency of Scheme 1 for Numerical Experiment 2

We use Taylor's series expansion about $(n, i)$ in order to determine its order of accuracy. We obtain

$$v_t + 2v_x + 5v_{xxx} = -\frac{k^2}{6}v_{ttt} - \frac{1}{6}h^2 v_{xxx} + \dots$$

and we conclude that the accuracy of the scheme is quadratic both in space and in time.

### 6.3. Numerical Results

In this section, the approximation of the linear dispersive Equation (15) by Scheme 1 (29) is presented. Within the stability region, the spatial and the temporal step sizes employed are $h = \pi/10$ and $k = 0.001$, respectively. The solution profiles are shown in Figure 18. These are compared with exact solution $u(x, t) = \sin(x + 3t)$. The profile of absolute errors is also presented at $T = 2.0$ and $T = 4.0$. Figure 19 displays the absolute error profiles of Scheme 1 for Numerical Experiment 2. We compare $L_1$ and $L_\infty$ errors in Table 4.

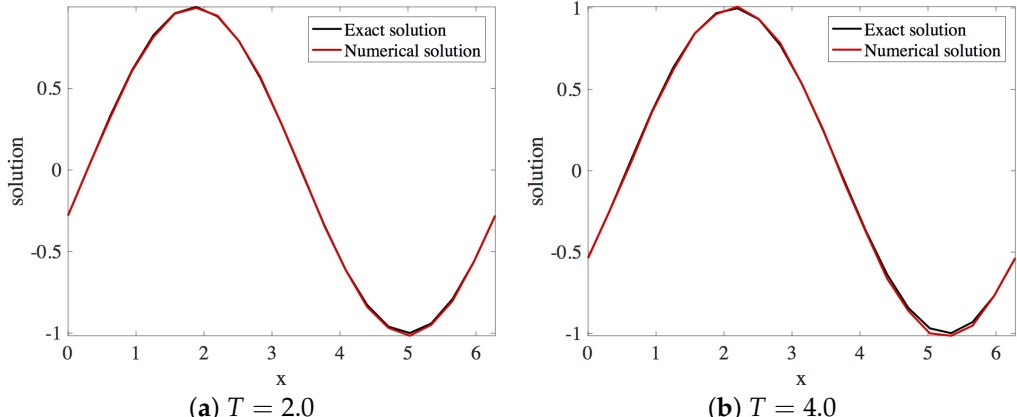

(a) $T = 2.0$      (b) $T = 4.0$

**Figure 18.** Plot of exact and numerical profiles (using Scheme 1) vs. $x$ at times 2.0 and 4.0 using $h = \pi/10$ and $k = 0.001$ (Numerical Experiment 2).

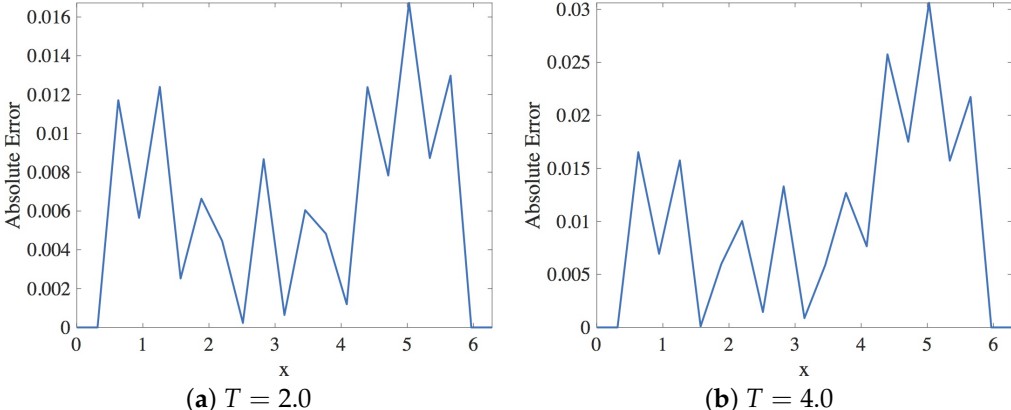

**(a)** $T = 2.0$        **(b)** $T = 4.0$

**Figure 19.** Plot of the absolute error vs. $x$ using Scheme 1 for Numerical Experiment 2. The spatial and temporal step sizes are $h = \pi/10$ and $k = 0.001$, respectively.

**Table 4.** $L_1$ and $L_\infty$ errors for various $k$ when $h = \pi/10$ when Scheme 1 is employed to approximate Numerical Experiment 2 at $T = 2$ and $T = 4$.

| | Error | | | |
|:---:|:---:|:---:|:---:|:---:|
| **Step Sizes $k$** | **$T = 2$** | | **$T = 4$** | |
| | $L_1$ $(\times 10^{-1})$ | $L_\infty$ $(\times 10^{-2})$ | $L_1$ | $L_\infty$ $(\times 10^{-1})$ |
| 0.001 | 1.2362 | 1.6724 | 2.0847 | 3.0605 |
| 0.002 | 1.2322 | 1.6849 | 2.0852 | 3.0651 |

*6.4. Dispersion Analysis*

We consider $u_t + 2u_x + 5u_{xxx} = 0$. We use same techniques as in Section 3.4 The amplification factor is determined from the relation

$$\xi_{exact} = \frac{u(x, t^{n+1})}{u(x, t^n)} = e^{I\theta(5\theta^2 - 2)k}.$$

The relative phase error (RPE) is defined [27]

$$RPE = \frac{arg(\xi_{numerical})}{arg(\xi_{exact})} = \frac{arg(\xi_{numerical})}{\theta(5\theta^2 - 2)k}.$$

We obtain plots of the arguments of $\xi_{exact}$, $\xi_1$, $\xi_2$ vs. $w \in [0, \pi]$ in Figure 20. For values of $w \in [0, 1]$ the graphs of the arguments of both $\xi_{exact}$ and $\xi_1$ are very close to each other, as illustrated in Figure 20a.

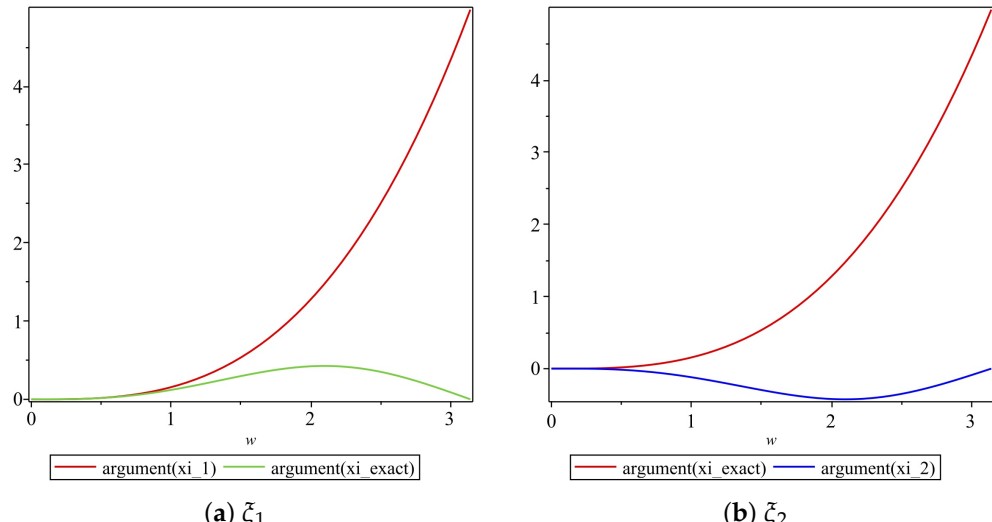

**(a)** $\xi_1$    **(b)** $\xi_2$

**Figure 20.** Plot of the arguments of amplification factors of Scheme 1 for Numerical Experiment 2 with the spatial and temporal step sizes $h = \pi/10$ and $k = 0.001$, respectively.

## 7. Scheme 2 for Numerical Experiment 2

Wang et al. [14] proposed

$$\frac{1}{2k}(v_i^{n+1} - v_i^n + v_{i+1}^n - v_{i+1}^{n-1}) + \frac{\eta}{2h}(v_{i+1}^n + v_i^n)(v_{i+1}^n - v_i^n) + \frac{\mu^2}{h^3}(v_{i+2}^n - 3v_{i+1}^n + 3v_i^n - v_{i-1}^n) = 0. \tag{30}$$

For the approximation of the nonlinear KdV equation $u_t + \eta u u_x + \delta^2 u_{xxx} = 0$, we adapt the method in Equation (30) to approximate Equation (15). The proposed method is

$$\frac{1}{2k}(v_i^{n+1} - v_i^n + v_{i+1}^n - v_{i+1}^{n-1}) + \frac{2}{h}(v_{i+1}^n - v_i^n) + \frac{5}{h^3}(v_{i+2}^n - 3v_{i+1}^n + 3v_i^n - v_{i-1}^n) = 0. \tag{31}$$

Analysis the stability, consistency and the dispersion properties of the Scheme given by ((31)) are discussed below.

### 7.1. Stability of Scheme 2 for Numerical Experiment 2

Scheme (31) can be rewritten explicitly as

$$v_i^{n+1} = v_i^n - v_{i+1}^n + v_{i+1}^{n-1} - \frac{4k}{h}(v_{i+1}^n - v_i^n) - \frac{10k}{h^3}(v_{i+2}^n - 3v_{i+1}^n + 3v_i^n - v_{i-1}^n) \tag{32}$$

for all $i = 2, 3, 4, \ldots, NP - 2$ and $n = 2, 3, \ldots, itmax - 1$. The amplification factor satisfies

$$\xi^2 + A\xi + B = 0,$$

where

$$A = \frac{4k}{h}(e^{I\theta h} - 1) + \frac{10k}{h^3}(e^{2I\theta h} - 3e^{I\theta h} + 3 - e^{-I\theta h}) - (1 - e^{I\theta h})$$

and

$$B = -e^{I\theta h}.$$

The roots of this equation are

$$\xi_{1,2} = \frac{1}{2}(-A \pm \sqrt{A^2 + 4B}).$$

We fix $h = \pi/10$ and obtain plots of $|\xi_1|$ and $|\xi_2|$, vs. $w \in [-\pi, \pi]$ vs. $k$ in Figures 21 and 22, respectively, and we deduce that the stability region is $0 < k < 0.001$.

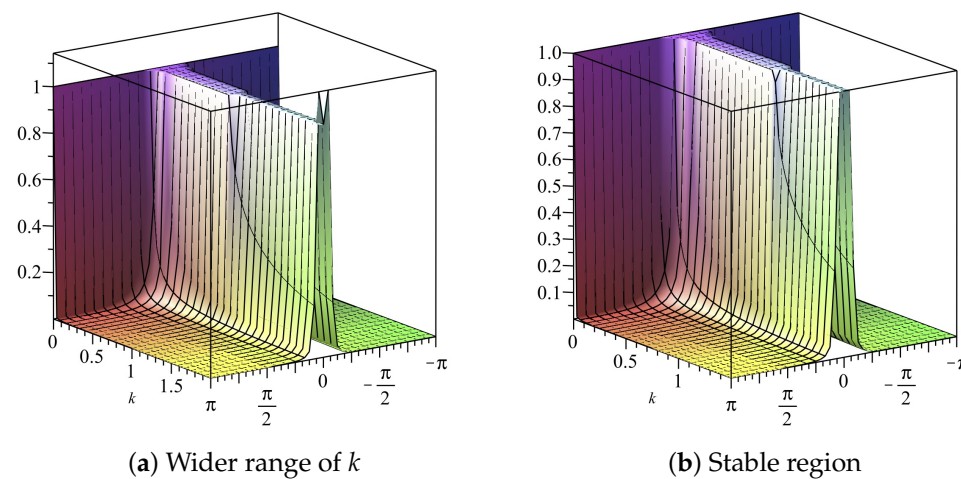

(**a**) Wider range of $k$        (**b**) Stable region

**Figure 21.** Plot of $|\xi_1|$ vs. $k$ vs. $w \in [-\pi, \pi]$ of Scheme 2 for Numerical Experiment 2. The spatial step size $h = \pi/10$.

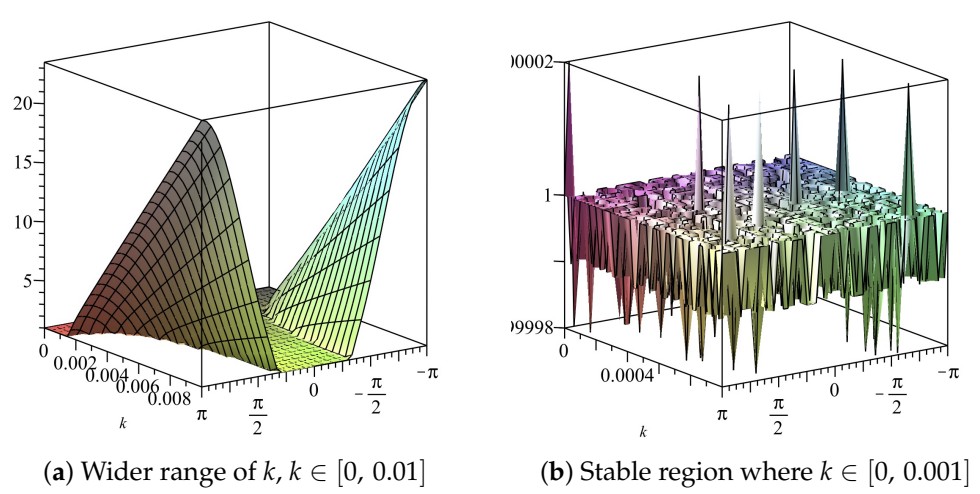

(**a**) Wider range of $k$, $k \in [0, 0.01]$        (**b**) Stable region where $k \in [0, 0.001]$

**Figure 22.** Plot of $|\xi_2|$ vs. $k$ vs. $w \in [-\pi, \pi]$ of Scheme 2 for Numerical Experiment 2. The spatial step size $h = \pi/10$.

*7.2. Consistency of Scheme 2 for Numerical Experiment 2*

In this section, the consistency of the numerical scheme (31) is discussed. We consider Equation (31) and use Taylor's series expansion about $(n, i)$ to get

$$v_t + 2v_x + 5v_{xxx} = -\frac{k^3}{6}v_{ttt} - \frac{h}{2}v_{tx} + \frac{kh}{4}v_{txx} - \frac{h^2}{12}v_{txx} - hv_{xx} - \frac{h^2}{3}v_{xxx} - \frac{h^3}{12}v_{xxxx} - \frac{5h}{2}v_{xxxx} + \dots$$

We note that

$$v_{tx} + 2v_{xx} + 5v_{xxxx} \approx 0$$

and we therefore obtain

$$v_t + 2v_x + 5v_{xxx} = -\frac{k^3}{6}v_{ttt} - \frac{h^2}{3}v_{xxx} + \frac{kh}{4}v_{txx} - \frac{h^2}{12}v_{txx} - \frac{h^3}{12}v_{xxxx} + \dots$$

We deduce that the scheme is of second order accuracy in both time and space.

### 7.3. Numerical Result

In this section, the approximation of the linear dispersive Equation (15) by Scheme 2, given in Equation (31), is presented. Within the stability region, the spatial and the temporal step sizes employed are $h = \pi/10$ and $k = 0.001$, respectively. The solution profiles are shown in Figure 23. These are compared with exact solution $u(x, t) = \sin(x + 3t)$. The profile of absolute errors are presented at $T = 2.0$ and $T = 4.0$. Figure 24 displays the absolute error profiles. We display $L_1$ and $L_\infty$ errors in Table 5.

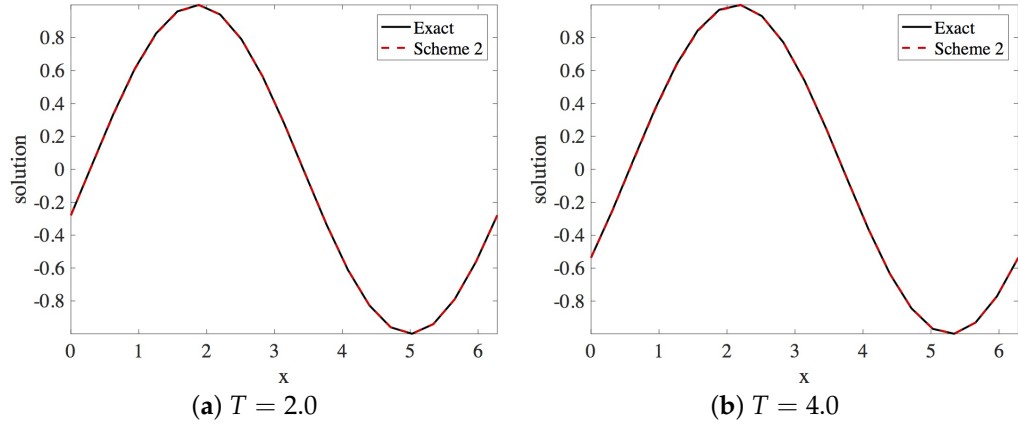

(**a**) $T = 2.0$      (**b**) $T = 4.0$

**Figure 23.** Plot of exact and numerical profiles (using Scheme 2) vs. $x$ at times 2.0 and 4.0 using $h = \pi/10$ and $k = 0.001$ (Numerical Experiment 2).

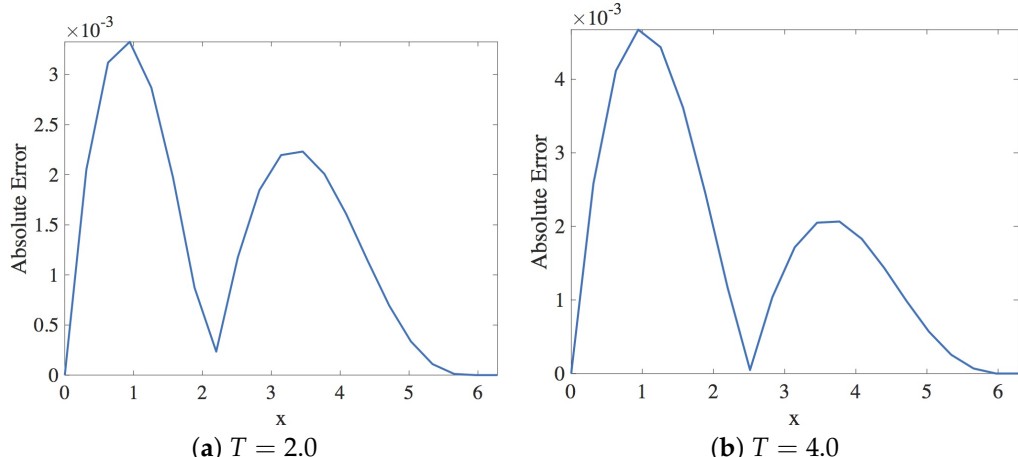

(**a**) $T = 2.0$      (**b**) $T = 4.0$

**Figure 24.** Plot of the absolute error vs. $x$ using Scheme 2 for Numerical Experiment 2. The spatial and temporal step sizes are $h = \pi/10$ and $k = 0.001$, respectively.

**Table 5.** $L_1$ and $L_\infty$ errors for various $k$ when $h = \pi/10$ when Scheme 2 is employed to approximate Numerical Experiment 2 at $T = 2$ and $T = 4$.

| Step Sizes $k$ | Error | | | |
| --- | --- | --- | --- | --- |
| | $T = 2$ | | $T = 4$ | |
| | $L_1$ ($\times 10^{-2}$) | $L_\infty$ ($\times 10^{-3}$) | $L_1$ ($\times 10^{-2}$) | $L_\infty$ ($\times 10^{-3}$) |
| 0.0001 | 2.6768 | 3.2056 | 3.3807 | 4.5001 |
| 0.0002 | 2.6885 | 3.2190 | 3.3954 | 4.5195 |
| 0.0003 | 2.6997 | 3.2306 | 3.4106 | 4.5405 |
| 0.0004 | 2.7119 | 3.2460 | 3.4245 | 4.5582 |
| 0.0005 | 2.7236 | 3.2595 | 3.4391 | 4.5776 |
| 0.0006 | 2.7362 | 3.2769 | 3.4523 | 4.5935 |
| 0.0007 | 2.7473 | 3.2883 | 3.4695 | 4.6195 |
| 0.0008 | 2.7584 | 3.2998 | 3.4827 | 4.6354 |
| 0.0009 | 2.7710 | 3.3172 | 3.4998 | 4.6614 |
| 0.001 | 2.7816 | 3.3266 | 3.5116 | 4.6739 |

*7.4. Dispersion Analysis*

We plot the arguments of the numerical amplification factors as derived in Section with the exact amplification factor in Figure 25. The argument of $\xi_1$ and the argument of exact amplification factor are quite close to each other for $w \in [0, 1]$.

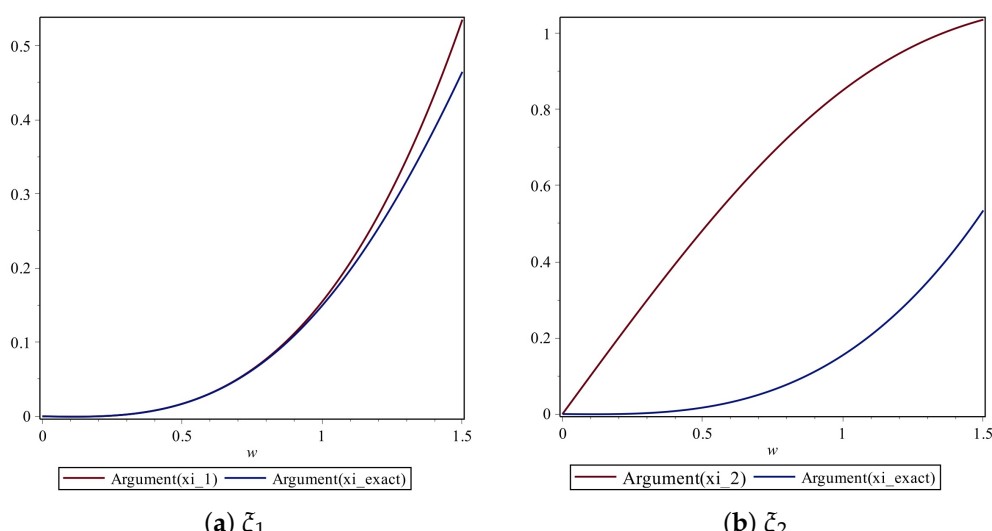

(**a**) $\xi_1$      (**b**) $\xi_2$

**Figure 25.** Plot of the arguments of amplification factors of Scheme 2 for Numerical Experiment 2 with the spatial and temporal step sizes $h = \pi/10$ and $k = 0.001$, respectively.

## 8. Scheme 3 for Numerical Experiment 2

Wang et al. [1] proposed the scheme

$$\frac{1}{2k}(v_{i-1}^{n+1} - v_{i-1}^n + v_{i+1}^n - v_{i+1}^{n-1}) + \eta \frac{v_{i-1}^n + v_i^n + v_{i+1}^n}{3} \frac{v_{i+1}^n - v_{i-1}^n}{2h}$$

$$+ \frac{\mu^2}{2h^3}(v_{i+2}^n - 2v_{i+1}^n + 2v_{i-1}^n - v_{i-2}^n) = 0 \tag{33}$$

for the numerical approximation of the nonlinear equation. We adapt this scheme to solve the linear dispersive equation $u_t + 2u_x + 5u_{xxx} = 0$. The scheme for Equation (15) is

$$\frac{1}{2k}(v_{i-1}^{n+1} - v_{i-1}^n + v_{i+1}^n - v_{i+1}^{n-1}) + 2\left(\frac{v_{i+1}^n - v_{i-1}^n}{2h}\right) + \frac{5}{2h^3}(v_{i+2}^n - 2v_{i+1}^n + 2v_{i-1}^n - v_{i-2}^n) = 0. \tag{34}$$

In explicit form, after some index shift, this equation is written as

$$v_i^{n+1} = v_i^n - v_{i+2}^n + v_{i+2}^{n-1} - \frac{2k}{h}(v_{i+2}^n - v_i^n) - \frac{5k}{h^3}(v_{i+3}^n - 2v_{i+2}^n + 2v_i^n - v_{i-1}^n). \tag{35}$$

### 8.1. Stability of Scheme 3 for Numerical Experiment 2

We substitute $v_i^n$ by $\xi^n e^{I\theta i h}$, and after simplification we get

$$\xi = 1 - e^{2I\theta h} + \xi^{-1}e^{2I\theta h} - \frac{2k}{h}(e^{2I\theta h} - 1) - \frac{5k}{h^3}(e^{3I\theta h} - 2e^{2I\theta h} + 2 - e^{-I\theta h}) + e^{2I\theta h}.$$

This is a quadratic equation in $\xi$, written as $\xi^2 + A\xi + B = 0$ where

$$A = \frac{2k}{h}(e^{2I\theta h} - 1) + \frac{5k}{h^3}(e^{3I\theta h} - 2e^{2I\theta h} + 2 - e^{-I\theta h}) - (1 - e^{2I\theta h})$$

and

$$B = -e^{2I\theta h}.$$

The amplification factors $\xi_{1,2}$ are, therefore, obtained as $\xi_1 = \frac{1}{2}(-A + \sqrt{A^2 + 4B})$ and $\xi_2 = \frac{1}{2}(-A - \sqrt{A^2 + 4B})$. The stability constraint for the scheme is computed from each of these amplification factors using $h = \frac{\pi}{10}$ for $\theta h = w \in [-\pi, \pi]$. The stability region is deduced from Figures 26 and 27 as $0 < k \le 0.001$.

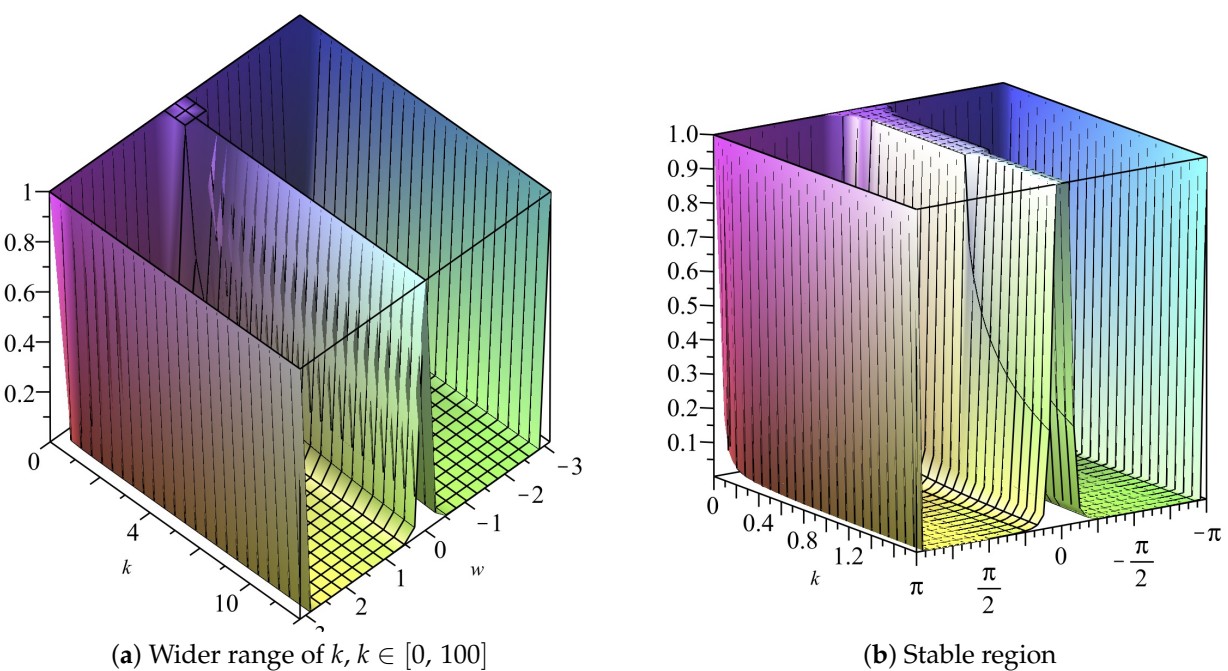

**(a)** Wider range of $k$, $k \in [0, 100]$  **(b)** Stable region

**Figure 26.** Plot of $|\xi_1|$ vs. $k$ vs. $w$ of Scheme 3 for Numerical Experiment 2. The spatial step size $h = \pi/10$.

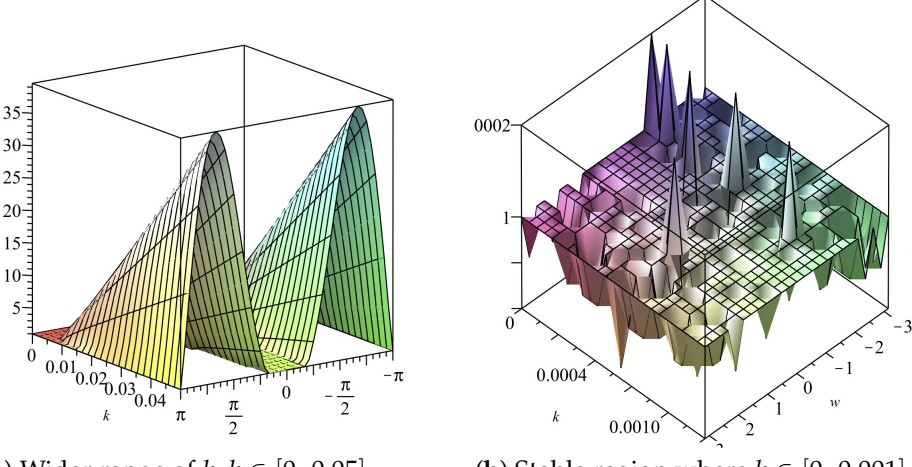

(**a**) Wider range of $k$, $k \in [0, 0.05]$      (**b**) Stable region where $k \in [0, 0.001]$

**Figure 27.** Plot of $|\xi_2|$ vs. $k$ vs. $w \in [-\pi, \pi]$ of Scheme 3 for Numerical Experiment 2. The spatial step size $h = \pi/10$.

### 8.2. Consistency of Scheme 3 for Numerical Experiment 2

To discuss the consistency of the scheme in Equation (35), consider the Taylor expansion about $(n, i)$. After simplification, we get

$$\frac{1}{2k}\left(2kv_t + 2hkv_{tx} + \frac{2k^3}{6}v_{ttt} + 2h^2kv_{txx} - hk^2v_{ttx}\right)$$

$$+\frac{1}{h}\left(2hv_x + \frac{4h^2}{2}v_{xx} + \frac{8h^3}{6}v_{xxx} + \frac{(2h)^4}{24}v_{xxxx}\ldots\right) + \frac{5}{2h^3}\left(2h^3v_{xxx} + 2h^4v_{xxxx} + \ldots\right) = 0.$$

We obtain

$$v_t + 2v_x + 5v_{xxx} = -\frac{k^2}{6}v_{ttt} - h^2v_{txx} + \frac{hk}{2}v_{ttx} - \frac{4}{3}h^2v_{xxx} - \frac{2}{3}h^3v_{xxxx} + \ldots,$$

hence, the scheme is spatially and temporally accurate of order 2.

### 8.3. Numerical Results

In this section, the approximate solution of the linear dispersive Equation (15) by Scheme 3, given in Equation (35), is presented. Within the stability region, the spatial and the temporal step sizes employed are $h = \pi/10$ and $k = 0.001$, respectively. The solution profiles are shown in Figure 28. These are compared with exact solution $u(x, t) = \sin(x + 3t)$. The profile of absolute errors is also presented at $T = 2.0$ and $T = 4.0$. Figure 29 displays the absolute error profiles for the multisymplectic scheme. We display $L_1$ and $L_\infty$ errors in Table 6.

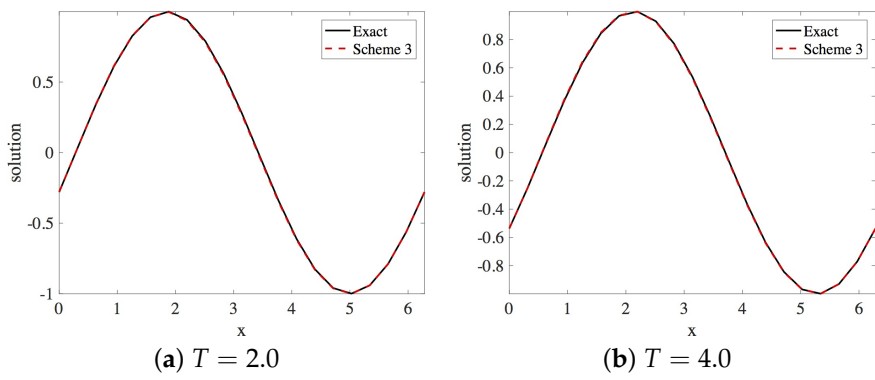

**(a)** $T = 2.0$      **(b)** $T = 4.0$

**Figure 28.** Plot of exact and numerical profiles (using Scheme 3) vs. $x$ at times 2.0 and 4.0 using $h = \pi/10$ and $k = 0.001$ (Numerical experiment 2).

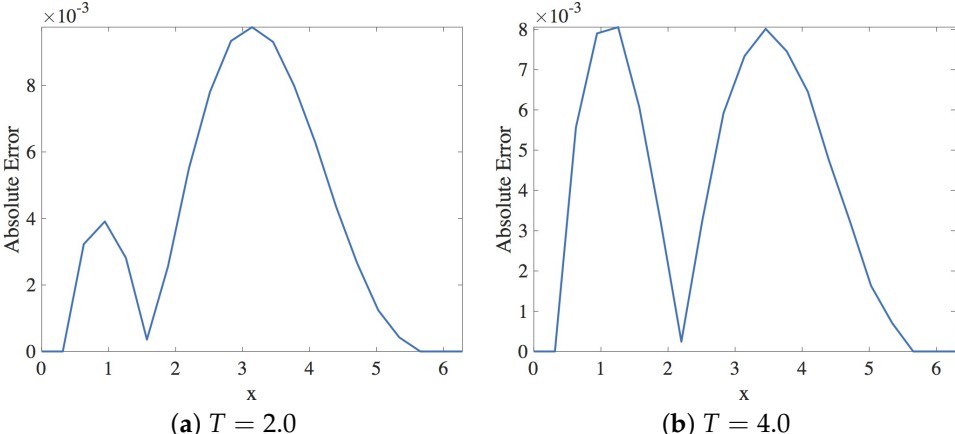

**(a)** $T = 2.0$      **(b)** $T = 4.0$

**Figure 29.** Plot of the absolute error vs. $x$ using Scheme 3 for Numerical Experiment 2. The spatial and temporal step sizes are $h = \pi/10$ and $k = 0.001$, respectively.

**Table 6.** $L_1$ and $L_\infty$ errors for various $k$ when $h = \pi/10$ when Scheme 3 is employed to approximate Numerical Experiment 2 at $T = 2$ and $T = 4$.

| | Error | | | |
|---|---|---|---|---|
| **Step Sizes $k$** | $T = 2$ | | $T = 4$ | |
| | $L_1\ (\times 10^{-2})$ | $L_\infty\ (\times 10^{-3})$ | $L_1\ (\times 10^{-2})$ | $L_\infty\ (\times 10^{-3})$ |
| 0.0001 | 7.8870 | 9.9051 | 8.1634 | 8.24321 |
| 0.0002 | 7.8720 | 9.8882 | 8.1440 | 8.2225 |
| 0.0003 | 7.8554 | 9.8643 | 8.1247 | 8.2082 |
| 0.0004 | 7.8422 | 9.8545 | 8.1254 | 8.1810 |
| 0.0005 | 7.8273 | 9.8377 | 8.0860 | 8.1602 |
| 0.0006 | 7.8107 | 9.8140 | 8.0669 | 8.1652 |
| 0.0007 | 7.7966 | 9.8007 | 8.0475 | 8.1316 |
| 0.0008 | 7.7826 | 9.7873 | 8.0282 | 8.0981 |
| 0.0009 | 7.7661 | 9.7638 | 8.0091 | 8.1029 |
| 0.001 | 7.7529 | 9.7538 | 7.9897 | 8.0567 |
| 0.002 | 7.6051 | 9.5873 | 7.7978 | 7.8503 |
| 0.003 | 7.4446 | 9.3599 | 7.6076 | 7.7051 |
| 0.004 | 8.0514 | 12.2661 | 7.6774 | 18.6819 |

### 8.4. Dispersion Analysis

The arguments of the numerical amplification factors $\xi_{1,2}$ are compared with the argument of the exact solution of the linear dispersive Equation (15). The amplification factors are derived in Section 8.1. The comparisons are shown in Figure 30.

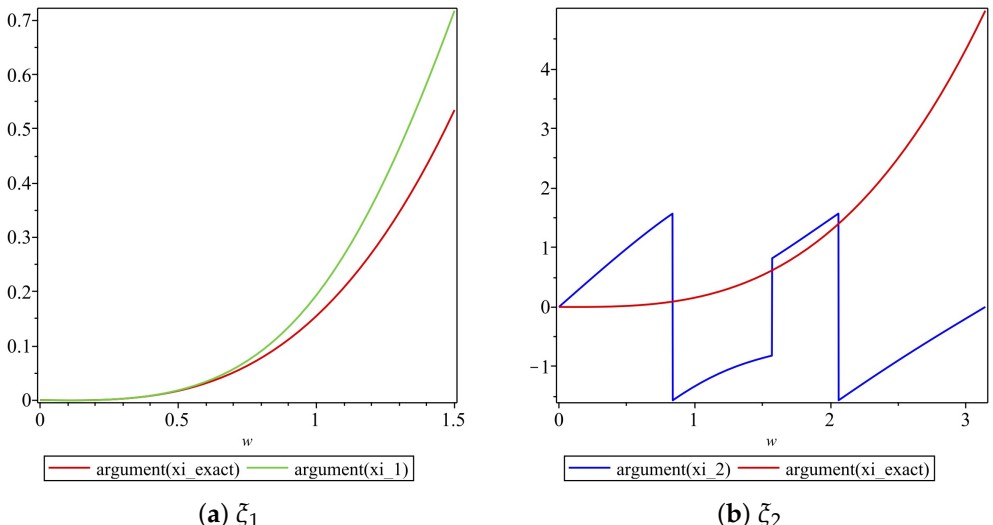

(a) $\xi_1$        (b) $\xi_2$

**Figure 30.** Plot of the arguments of amplification factors of Scheme 3 for Numerical Experiment 2 with the spatial and temporal step sizes are $h = \pi/10$ and $k = 0.001$, respectively.

### 9. Conclusions

This paper considers, proposes and analyzes three finite difference schemes for two linear dispersive KdV equations. One of the cases is such that the advective term dominates the dispersive term, while in the second case, the dispersive term dominates the advective term. Stability regions were derived for the schemes, their accuracies were discussed and their dispersion analyses were conducted and presented.

It is observed that for Numerical Experiment 1, the stability region for Scheme 3, $0 < k \leq 0.001$, is narrower compared to the stability region $0 < k \leq 0.01$ for Scheme 1 and Scheme 2. The maximum absolute error of Scheme 1 is relatively less than the maximum absolute error for Scheme 2 and Scheme 3 both for short time and long time integration. In addition, the stability restraint is relaxed when Scheme 1 is employed to approximate Numerical Experiment 2 than Scheme 2 and Scheme 3. In this case, the maximum absolute error, when Scheme 2 is employed to solve Numerical Experiment 2 is less compared to Scheme 1 and Scheme 3. In fact, the highest error is recorded for Scheme 1.

A comparison of results of classical finite difference methods with Laplace Adomian Decomposition Method for Numerical Experiment 1 was done in [30], and it was observed that at a low propagation time of 0.1 s, LADM is better than the classical scheme but that at long time propagation (say when time = 4.0), the performance of LADM deteriorates significantly.

We intend to use the knowledge acquired in this paper to construct methods for linearized KdV equations in both $1 - D$ and $2 - D$ with more challenging initial conditions and check which of the three schemes is the best performing. Moreover, we can also construct similar methods for PDEs quite related to the KdV equations, especially those with $uu_x$ and $u_{xxx}$ terms present and these examples of PDEs are fractional KdV, modified KdV, KdV–Burgers–Kuramoto, KdV-Burgers and stochastic KdV equations.

**Author Contributions:** Conceptualization, A.R.A.; methodology, A.A.A. and A.R.A.; software, A.A.A. and A.R.A.; validation, A.A.A., A.R.A.; formal analysis, A.A.A. and A.R.A.; investigation, A.A.A. and A.R.A.; resources, A.A.A. and A.R.A.; writing—original draft preparation, A.R.A.; typing, A.A.A.; writing—review and editing, A.A.A. and A.R.A.; supervision, A.R.A.; project administration,

A.R.A.; funding acquisition, A.R.A. All authors have read and agreed to the published version of the manuscript.

**Funding:** This research received no external funding.

**Data Availability Statement:** Not applicable.

**Acknowledgments:** The authors are very grateful to the four anonymous reviewers who provided feedback and suggestions which enabled us to improve the paper considerably.

**Conflicts of Interest:** The authors declare no conflict of interest.

## Abbreviations

The following abbreviations are used in this manuscript:

KdV    Korteweg–de-Vries
PDE    Partial differential equation

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
