# Peer review of "Classical and Multisymplectic Schemes for Linearized KdV Equation: Numerical Results and Dispersion Analysis"

_fluids, doi:10.3390/fluids6060214_

Round 1
Reviewer 1 Report
This paper is interesting and is helpful for user to choose and to judge the methods used for simulation.
It can be published after some typos corrections, such as the equation must be numbered one by one.
Author Response
General comments to Reviewer 1
The authors have read and studied the four reports from all the four reviewers. The authors have revised the paper by implementing all the suggestions and modifications required from the reviewers. The main changes to the paper are as follows: The title has been changed slightly following suggestion from Reviewer 3. Some sections were shortened and two references added following feedback from Reviewer 4. Future work is given following comments from Reviewer 2. Most equations are now numbered as advised by Reviewer 1. The main highlights of the paper are: 1. This is one of the rare papers whereby dispersion analysis is done on numerical methods for dispersive partial differential equations (pdes). We are able to do an accurate dispersion analysis as we considered a linear pde. 2. Detailed analysis of consistency and stability in regards to both physical and computational modes have been done. The study is much more detailed as compared to work from Wang et al. [1, 14].
Specific comments to Reviewer 1
The authors will like to thank Reviewer 1 for the comments and suggestions. The paper was read many times and modified after considering all points raised by the 4 reviewers. Then, the paper has been carefully checked for typo and most equations are now labelled.
Reviewer 2 Report
The authors have shown an original work. The points to be amended are mentioned below:
(1) In conclusion section, the last paragraph starts with "in addition". It would be better to include this part as a part of the previous paragraph instead of the beginning of a new paragraph.
(2) At the end of conclusion, it would be useful to mention the future perspectives related to the studies carried out.
Author Response
General comments to Reviewer 2
The authors have read and studied the four reports from all the four reviewers. The authors have revised the paper by implementing all the suggestions and modifications required from the reviewers.
The main changes to the paper are as follows:
The title has been changed slightly following suggestion from Reviewer 3. Some sections were shortened and two references added following feedback from Reviewer 4.
Future work is given following comments from Reviewer 2.
Most equations are now numbered as advised by Reviewer 1.
The main highlights of the paper are:
- This is one of the rare papers whereby dispersion analysis is done on numerical methods for dispersive partial differential equations (pdes). We are able to do an accurate dispersion analysis as we considered a linear pde.
- Detailed analysis of consistency and stability in regards to both physical and computational modes have been done. The study is much more detailed as compared to work from Wang et al. [1, 14].
Specific comments to reviewer 2
The authors would like to thank Reviewer 2 for the constructive comments and suggestions. In the conclusion, the paragraph which starts with ‘In addition’ has been merged with the previous paragraph.
Also, the future perspectives of this work have now been provided at the end of the conclusion. We intend to use the knowledge acquired in this paper to construct methods for linearized KdV equations with a more challenging initial conditions and check which of the three schemes is best performing. Moreover, we can also construct similar methods for pdes quite related to the KdV equations, especially those with uux and uxxx terms present and these examples of pdes are fractional KdV, modified KdV, KdV-Burgers-Kuramoto, KdV Burgers and stochastic KdV equations.
Reviewer 3 Report
The authors present numerical solutions of two linearised KDV equations with different coefficients using three different numerical methods: the original scheme by Zabusky and Kruskal, the 6-point multisymplectic scheme and the modified explicit scheme proposed by Wang et al (2007). The authors apply all three schemes to both linear equations, obtaining stability properties and comparing absolute errors for a chosen temporal and spatial resolution.
Although the paper does not have much focus (in fact, the authors just repeat the same steps six times overall, and perform some basic comparisons), it may be of some interest, as a comparison of performance of symplectic and classical algorithms. However, due to the fact that the considered equations are very simple, with sinusoidal exact solutions, it is difficult to make meaningful conclusions from the authors' results. The equations under consideration are chosen arbitrarily, and it is not clear to what extent their conclusions are relevant in a broader context. Moreover, a more meaningful comparison of the same numerical schemes in the context of the full KdV equations has already been performed by Wang at al.
I find it hard to understand why the authors have chosen to submit their paper to Fluids in the first place. They are not considering any fluid dynamics problem, or any physical problem at all. They may consider submitting their work to some numerical analysis journal, although even in this case the paper needs more focus and more emphasis on particular insights arising from the authors' contribution. As the manuscript stands, I cannot recommend publication in Fluids.
Author Response
General comments to Reviewer 3
The authors have read and studied the four reports from all the four reviewers. The authors have revised the paper by implementing all the suggestions and modifications required from the reviewers.
The main changes to the paper are as follows:
The title has been changed slightly following suggestion from Reviewer 3. Some sections were shortened and two references added following feedback from Reviewer 4.
Future work is given following comments from Reviewer 2.
Most equations are now numbered as advised by Reviewer 1.
The main highlights of the paper are:
- This is one of the rare papers whereby dispersion analysis is done on numerical methods for dispersive partial differential equations (pdes). We are able to do an accurate dispersion analysis as we considered a linear pde.
- Detailed analysis of consistency and stability in regards to both physical and computational modes have been done. The study is much more detailed as compared to work from Wang et al. [1, 14].
Specific comments to Reviewer 3
The authors are grateful to Reviewer 3 for the comments and suggestions. The authors have considered all the points raised by Reviewer 3 and revised the paper accordingly.
The research design has been improved and results are better presented. The concluding section has been improved.
The work of the authors is quite different from Wang et al. (2008). Wang et al. (2008) considered three schemes for u_t+eta u u_x+delta^2 u_xxx with initial condition u(x,0)=cos(pi x) and three schemes are used namely: Zabusky-Kruskal, multi-symplectic six -point scheme and a modified explicit scheme. Firstly, no consistency analysis was done. Secondly, using von Neumann stability analysis and freezing coefficient technique, Wang et al. (2008) obtained stability region of the three schemes and they found out that the stability region of the multi-symplectic six-point scheme and the modified explicit scheme is larger than that of the Zabusky-Kruskal scheme. We note that the freezing coefficient technique is an approximate method. Thirdly, Wang et al. did not give expressions for the amplification factor of the physical and computational modes. Fourthly, there is no exact solution and therefore Wang et al. (2008) had to compare the errors of the discrete conservation laws with time. The main findings of Wang et al. (2008) is that their new scheme is more stable than the multi-symplectic six-point scheme and that the new scheme is still stable at time t = 40 while the Zabusky-Kruskal scheme experience blow up at t>6.5. So, their new scheme is stable at long time propagation.
In our work, we chose a linearized KdV equation with specified initial and boundary conditions for which an exact solution is known so that we can compare the absolute errors. We chose a linearize KdV equation so that we can get true and not approximate expressions for the amplification factor of the physical and computational modes in order to obtain exact stability region and do a dispersion analysis. We did a detailed study of the consistency and stability of the schemes. We chose a linearized KdV equation to check how performance of the methods can change as compared to results from Wang et al. (2008). The numerical experiment we considered is quite popular and was solved using Adomian decomposition method, Homotopy perturbation method, Bernstein-Laplace-Adomian method and Reduced Differential Transform method. We also considered a new case (case 2) with pde being u_t+2 u_x+5 u_xxx=0 to check performance of schemes when coefficient of dispersion is greater than coefficient of advection.
In concluding section, we mentioned that our future work will be based on knowledge acquired from this work.
The authors submitted their work to Fluids as the KdV equation is associated with fluids. The nonlinear KdV equation describes the long time asymptotic behavior of small but finite amplitude of one dimensional shallow water waves where u = u(x, t) measures the elevation at time t and position x while delta2 represents the dispersion coefficient.
Reviewer 4 Report
All comments and suggestions are given in the attached file

Author Response
General comments to Reviewer 4
The authors have read and studied the four reports from all the four reviewers. The authors have revised the paper by implementing all the suggestions and modifications required from the reviewers.
The main changes to the paper are as follows:
The title has been changed slightly following suggestion from Reviewer 3. Some sections were shortened and two references added following feedback from Reviewer 4.
Future work is given following comments from Reviewer 2.
Most equations are now numbered as advised by Reviewer 1.
The main highlights of the paper are:
- This is one of the rare papers whereby dispersion analysis is done on numerical methods for dispersive partial differential equations (pdes). We are able to do an accurate dispersion analysis as we considered a linear pde.
- Detailed analysis of consistency and stability in regards to both physical and computational modes have been done. The study is much more detailed as compared to work from Wang et al. [1, 14].
Specific comments to reviewer 4
The authors are very grateful to Reviewer 4 for the points mentioned and the authors have implemented all these points and revised the paper accordingly.
- delta2 in Eqn. (1) is defined as the dispersion coefficient. In Section 5, eta represents coefficient of the non linear term u u_x.
- Sections 2 and 3 have been merged with the introduction as advised.
- Sections 4 and 5 are separate.
- delta2 is the coefficient of dispersion.
- The paper was revised after implementing all the requested changes from the four reviewers and then the paper was checked for typos.
- The authors have skipped some of the repeated details especially those parts which deal with consistency and stability. This has made the reading more smooth and reduced the number of pages slightly.
- The conclusion has been rewritten by mentioning few lines about performance of the schemes for the two cases. Scheme 1 is best method for case 1 followed by schemes 2, 3. Scheme 2 is most efficient scheme for case 2 followed by Scheme 3 and 1. In general, Scheme 2 is most efficient method.
Comparison of results of classical finite difference methods with Laplace Adomian Decomposition Method for Numerical Experiment 1 was done by Ndala and it was observed that at low propagation time of 0.1 s, LADM is better than the classical scheme but that at long time propagation (say when time = 4.0), the performance of LADM deteriorates significantly.
References
Y. I. Ndala. On the solution of linearized KdV Equations using semi-analytic and finite difference methods. Project submitted as partial fulfilment of the requirements for the degree of BSc (Hons) in Applied Mathematics at the Nelson Mandela University, Port Elizabeth, South Africa, Project submitted in Jan. 2021.
- The introduction has been modified to include more recent and relevant papers and these two papers are added to the list of references.
Round 2
Reviewer 3 Report
I have read the revised manuscript, which has been improved. I still have certain reservations about the significance of this study, but the authors have certainly done a good work on the problem they had stated. If the editors of Fluids and other reviewers think that this manuscript deserves a publication in Fluids, I have no objection either.
Reviewer 4 Report
The authors answered all my comments. The revised version is ready for publication.